# Discordant rearrangement of primary and anamnestic CD8+ T cell responses to influenza A viral epitopes upon exposure to bacterial superantigens: Implications for prophylactic vaccination, heterosubtypic immunity and superinfections

Courtney E. Meilleur[1], Arash Memarnejadian[1], Adil N. Shivji[1], Jenna M. Benoit[1], Stephen W. Tuffs[1], Tina S. Mele[2,3], Bhagirath Singh[1,4,5], Jimmy D. Dikeakos[1], David J. Topham[6], Hong-Hua Mu[7], Jack R. Bennink[8], John K. McCormick[1,4,5], S. M. Mansour Haeryfar[1,2,4,5,9]*

1 Department of Microbiology and Immunology, Western University, London, Ontario, Canada, 2 Division of General Surgery, Department of Surgery, Western University, London, Ontario, Canada, 3 Division of Critical Care Medicine, Department of Medicine, Western University, London, Ontario, Canada, 4 Lawson Health Research Institute, London, Ontario, Canada, 5 Centre for Human Immunology, Western University, London, Ontario, Canada, 6 David H. Smith Center for Vaccine Biology and Immunology, Department of Microbiology and Immunology, University of Rochester Medical Center, Rochester, New York, United States of America, 7 Division of Rheumatology, Department of Internal Medicine, University of Utah School of Medicine, Salt Lake City, Utah, United States of America, 8 Viral Immunology Section, Division of Intramural Research, National Institute of Allergy and Infectious Diseases, National Institutes of Health, Bethesda, Maryland, United States of America, 9 Division of Clinical Immunology & Allergy, Department of Medicine, Western University, London, Ontario, Canada

* Mansour.Haeryfar@schulich.uwo.ca

## Abstract

Infection with (SAg)-producing bacteria may precede or follow infection with or vaccination against influenza A viruses (IAVs). However, how SAgs alter the breadth of IAV-specific CD8+ T cell ($T_{CD8}$) responses is unknown. Moreover, whether recall responses mediating heterosubtypic immunity to IAVs are manipulated by SAgs remains unexplored. We employed wild-type (WT) and mutant bacterial SAgs, SAg-sufficient/deficient *Staphylococcus aureus* strains, and WT, mouse-adapted and reassortant IAV strains in multiple *in vivo* settings to address the above questions. Contrary to the popular view that SAgs delete or anergize T cells, systemic administration of staphylococcal enterotoxin B (SEB) or *Mycoplasma arthritidis* mitogen before intraperitoneal IAV immunization enlarged the clonal size of 'select' IAV-specific $T_{CD8}$ and reshuffled the hierarchical pattern of primary $T_{CD8}$ responses. This was mechanistically linked to the TCR Vβ makeup of the impacted clones rather than their immunodominance status. Importantly, SAg-expanded $T_{CD8}$ retained their IFN-γ production and cognate cytolytic capacities. The enhancing effect of SEB on immunodominant $T_{CD8}$ was also evident in primary responses to vaccination with heat-inactivated and live attenuated IAV strains administered intramuscularly and intranasally, respectively. Interestingly, in prime-boost immunization settings, the outcome of SEB administration

**Data Availability Statement:** All relevant data are within the manuscript and its Supporting Information files.

**Funding:** This work was supported by a Canadian Institutes of Health Research (CIHR) (http://www.cihr-irsc.gc.ca/) project grant (PJT – 156295) to S. M.M.H., and by the Intramural Research Program of the National Institutes of Health/National Institute of Allergy and Infectious Diseases (https://www.niaid.nih.gov/). C.E.M. was a recipient of an Alexander Graham Bell Canada Graduate Scholarship from Natural Sciences and Engineering Research Council of Canada (NSERC) (http://www.nserc-crsng.gc.ca/index_eng.asp). The funders had no role in study design, data collection and analysis, decision to publish, or preparation of the manuscript.

**Competing interests:** The authors have declared that no competing interests exist.

depended strictly upon the time point at which this SAg was introduced. Accordingly, SEB injection before priming raised CD127$^{high}$KLRG1$^{low}$ memory precursor frequencies and augmented the anamnestic responses of SEB-binding $T_{CD8}$. By comparison, introducing SEB before boosting diminished recall responses to IAV-derived epitopes drastically and indiscriminately. This was accompanied by lower Ki67 and higher Fas, LAG-3 and PD-1 levels consistent with a pro-apoptotic and/or exhausted phenotype. Therefore, SAgs can have contrasting impacts on anti-IAV immunity depending on the naïve/memory status and the TCR composition of exposed $T_{CD8}$. Finally, local administration of SEB or infection with SEB-producing *S. aureus* enhanced pulmonary $T_{CD8}$ responses to IAV. Our findings have clear implications for superinfections and prophylactic vaccination.

## Author summary

Exposure to bacterial superantigens (SAgs) is often a consequence of infection with common Gram-positive bacteria causing septic and toxic shock or food poisoning. How SAgs affect the magnitude, breadth and quality of infection/vaccine-elicited CD8$^+$ T cell ($T_{CD8}$) responses to respiratory viral pathogens, including influenza A viruses (IAVs), is far from clear. Also importantly, superinfections with IAVs and SAg-producing bacteria are serious clinical occurrences during seasonal and pandemic flu and require urgent attention. We demonstrate that two structurally distinct SAgs, including staphylococcal enterotoxin B (SEB), unexpectedly enhance primary $T_{CD8}$ responses to 'select' IAV-derived epitopes depending on the TCR makeup of the responding clones. Intriguingly, the timing of exposure to SEB dictates the outcome of prime-boost immunization. Seeing a SAg before priming raises memory precursor frequencies and augments anamnestic $T_{CD8}$ responses. Conversely, a SAg encounter before boosting renders $T_{CD8}$ prone to death or exhaustion and impedes recall responses, thus likely compromising heterosubtypic immunity to IAVs. Finally, local exposure to SEB increases the pulmonary response of immunodominant IAV-specific $T_{CD8}$. These findings shed new light on how bacterial infections and SAgs influence the effectiveness of anti-IAV $T_{CD8}$ responses, and have, as such, wide-ranging implications for preventative vaccination and infection control.

## Introduction

Co-infections and superinfections with bacterial and viral pathogens represent a significant challenge in myriad life-threatening conditions and illnesses, including toxic shock syndrome [1,2], sepsis [3,4], and influenza A virus (IAV) epidemics and pandemics [5–9]. Some of the bacterial culprits, such as *Staphylococcus aureus* and *Streptococcus pyogenes*, secrete exotoxins that function as superantigens (SAgs) by virtue of their ability to bind the lateral surface of major histocompatibility complex (MHC) II molecules and polyclonally activate MHC-restricted CD4$^+$ ($T_{CD4}$) and CD8$^+$ T cells ($T_{CD8}$) bearing select T cell receptor (TCR) Vβ families [10].

We previously demonstrated that Gram-positive bacterial SAgs can also activate invariant natural killer T (*i*NKT) and mucosa-associated invariant T (MAIT) cells [11,12]. This triggers a rapid and heavy-handed inflammatory response, typified by a 'cytokine storm', with rebound immunosuppression and susceptibility to potentially fatal secondary infections. Severe IAV

infections can on their own cause a cytokine storm in a strain-dependent manner. This creates a complex scenario involving numerous inflammatory mediators with pathogenic or protective roles in host defense against IAV and secondary bacterial infections [13].

$T_{CD8}$ play a pivotal part in clearing IAV infections. Naïve IAV-specific $T_{CD8}$ are primed by professional antigen (Ag)-presenting cells (APCs) displaying cognate peptide:MHC I complexes along with costimulatory signals [14]. They then proliferate and differentiate into fully armed cytotoxic T lymphocytes (CTLs) that utilize cytolytic effector molecules (*e.g.*, perforin and granzymes) to destroy infected cells as well as inflammatory cytokines that limit viral replication and transmission [15].

How bacterial SAgs manipulate IAV infection- and vaccination-elicited $T_{CD8}$ responses is poorly understood. Exposure to SAgs can create physical and/or functional holes in mouse T cell repertoires by eliminating or anergizing many T cells [16–18]. This is assumed to put a fraction of $T_{CD8}$ with antiviral specificities out of commission, at least temporarily. Paradoxically, we demonstrated that SAgs activate and expand preexisting IAV-specific memory $T_{CD8}$ in mice and humans [19], and not the opposite as conventionally predicted.

IAV has been a workhorse for studies of antiviral $T_{CD8}$ immunodominance, the marked tendency of T cells to respond to viral peptides in a predictable hierarchy based on the MHC class I allomorphs present in an individual. $T_{CD8}$ immunodominance hierarchies are controlled by many factors, including the type of APCs participating in Ag processing [20], the expression kinetics of viral proteins and the efficiency of their degradation in the cytosol [21,22], the rough selectivity of TAP for the resulting peptides [23], the affinity of MHC I molecules for transported peptides [23,24], and of course the frequencies of naïve peptide-specific $T_{CD8}$ in one's repertoire [25] that may indeed be altered by SAgs. We previously demonstrated that the breadth of $T_{CD8}$ responses can also be influenced by the immunomodulatory functions of $CD4^+CD25^+$ regulatory T (Treg) cells [26], intracellular indoleamine 2,3-dioxygenase (IDO) [27] and the cell surface checkpoint molecule programmed cell death-1 (PD-1) [28], and by the enzymatic actions of terminal deoxynucleotidyl transferase (TdT) [29,30] and the mammalian target of rapamycin (mTOR) [31].

Immunodominance is an intriguing phenomenon from a basic biological standpoint and is also considered an obstacle to successful vaccination [32]. *Bona fide* $T_{CD8}$ responses must be both sufficient in vigor and broad in scope to optimally target multiple peptide Ags. An immunodominant status does not necessarily confer a protective property upon any given T cell clone(s) [33] as subdominant $T_{CD8}$ responses can also contribute to anti-pathogen immunity.

The impact of bacterial SAgs on $T_{CD8}$ immunodominance is unknown. This represents a gap in our understanding of host responses to vaccination or infection with IAVs and also to other viruses. Furthermore, how recall $T_{CD8}$ responses to IAV may be manipulated by SAgs is unclear. Finally, whether the time point at which a SAg is encountered may change the immunization outcome is essentially unexplored. We have addressed the above questions by using multiple SAgs or SAg-sufficient/deficient bacteria in several *in vivo* settings in which $T_{CD8}$ responses could be monitored against wild-type (WT) and reassortant IAV strains, a live attenuated nasal flu vaccine, and vaccinia virus (VacV). The implications of our findings for sequential infections, superinfections, anti-IAV vaccination and heterosubtypic immunity will be discussed.

## Materials and methods

### Ethics statement

All animal experiments were carried out using age-matched mice under protocols approved by Animal Care and Veterinary Services at Western University (AUPs 2010–239, 2010–241 and 2018–093) and following the Canadian Council on Animal Care guidelines.

## Mice

Adult female BALB/c mice were purchased from Charles River Canada (Saint-Constant, QC) and cared for in our institutional barrier facility.

## WT and recombinant viruses

The mouse-adapted IAV strain A/Puerto Rico/8/1934 (PR8, H1N1) was used in the vast majority of experiments. Sequential 12 (SEQ12) is an escape mutant selected after sequential neutralization of PR8 in the presence of 12 different hemagglutinin (HA)-specific monoclonal antibodies (mAbs) [34]. Other IAV strains used in this study were A/Northern Territory/60/1968 (NT60, H3N2), A/Hong Kong/1/1968 (HK, H3N2), the X31 reassortant harboring a PR8 core and an HK coat (H3N2) [35], and the reassortant J-1 containing seven genes from PR8 plus an HK-derived HA (H3N1) [36]. All IAV strains were grown in 10-day-old embryonated chicken eggs. Infectious allantoic fluid was harvested, pooled, filtered, titrated and stored at -80˚C until use.

The 2015–2016 live attenuated nasal spray flu vaccine (FluMist, MedImmune, Gaithersburg, MD) containing a reassortant of the pandemic A/California/7/2009 (H1N1) strain was stored at 4˚C before use.

The Western Reserve strain of VacV was propagated in the thymidine kinase-deficient osteosarcoma cell line 143B.

## WT and mutant Bacteria

The methicillin-resistant *Staphylococcus aureus* COL was used in superinfection experiments. A mutant strain of this bacterium with a deletion of *seb*, the gene encoding staphylococcal enterotoxin B (SEB), was previously described [37]. These strains will be simply referred to as COL and COL Δ*seb*, respectively. For bacterial growth curve analysis, cells were picked from a tryptic soy agar (TSA) plate, grown overnight in Brain Heart Infusion (BHI) broth, and subcultured in half-BHI medium (BHI broth diluted 1:2 in sterile Milli-Q water). $OD_{600}$ was adjusted to 0.05 before 200 μL/well of cultures were placed in quadruplicate in a 96-well flat-bottom Microfluor 2 White plate (Thermo Fisher Scientific). Bacterial growth was monitored at 37˚C with shaking using a Biotek Synergy H4 plate reader, and $OD_{600}$ readings were recorded at one-hour intervals.

## Native and recombinant SAgs

Native *Mycoplasma arthritidis* mitogen (MAM) was purified as described elsewhere [38]. Briefly, *M. arthritidis* strain PG6 (ATCC 19611) was grown to senescence in autoclaved modified Edward-Hayflick medium, which was then subjected to $(NH_4)_2SO_4$ fractionation. This was followed by sequential rounds of gel filtration and cation-exchange chromatography prior to the final purification of MAM by fast protein liquid chromatography.

Recombinant staphylococcal enterotoxins A and B (SEA and SEB) were cloned from *S. aureus* strains Newman and COL, respectively, expressed in *Escherichia coli* BL21 (DE3), and then purified by nickel column chromatography as we previously described [39]. We also generated an SEB variant carrying an N → A point mutation at position 23 ($SEB_{N23A}$). N23 is critical for binding to mouse TCR Vβ8.2 and Vβ8.3 [11,40,41], prototypic targets of intact SEB.

SAg preparations were confirmed not to contain biologically meaningful endotoxin levels [12,42,43].

## SAg administration, immunization and infection protocols

A bolus dose of SEA, SEB or $SEB_{N23A}$ (50 μg in 200 μL PBS) was administered intraperitoneally (i.p.) at indicated time points. In a limited number of experiments, mice were briefly anesthetized using 20% isoflurane before they were injected intranasally (i.n.) with 50 ng SEB in 25 μL sterile PBS. Control mice received an equal volume of PBS. MAM was injected i.p. at 50 ng in 200 μL of PBS containing 0.2% BSA.

In our primary IAV immunization protocols, infectious allantoic fluid containing indicated viruses was diluted in sterile PBS before ~600 hemagglutinating units were injected i.p. into each mouse. In several experiments, PR8 was incubated in a 55˚C water bath for 30 minutes [44]. $TCID_{50}$ assays confirmed the loss of infectivity by heat-inactivated PR8, which was then injected intramuscularly (i.m.), 50 μL into each thigh muscle, using a 25-gauge needle.

Indicated cohorts of mice were vaccinated i.n. using 20 μL of FluMist under light isoflurane anesthesia.

To study recall $T_{CD8}$ responses to IAV, mice were first primed with PR8 followed, 30 days later, by a boosting i.p. injection of SEQ12 as we previously reported [29].

To induce active PR8 infection, a 25-μL inoculum of infectious allantoic fluid approximating 0.3 $MLD_{50}$ (50% mouse lethal dose) was instilled into the nares of anesthetized mice. Animals were monitored for signs of morbidity, including lethargy, piloerection and weight loss. On day 4 post-PR8 infection [45], the abundance of replicative viral particles was evaluated in lung homogenate specimens and in bronchoalveolar lavage (BAL) fluid and reported as 50% tissue culture infectious dose ($TCID_{50}$) using Madin-Darby Canine Kidney (MDCK) cells (ATCC CCL-34). We employed a previously published protocol in our $TCID_{50}$ assays [46] but utilized chicken red blood cell hemagglutination, as opposed to cytopathic effects, as a readout [47].

For superinfection experiments, COL and COL Δ*seb* colonies were picked from TSA plates, grown overnight in tryptic soy broth (TSB), and subcultured 2% (V/V) in TSB to yield an $OD_{600}$ of 3.0–3.5. Bacterial pellets were washed and adjusted to a concentration of $4 \times 10^9$ colony-forming units (CFUs)/mL in Hank's Balanced Salt Solution. Under isoflurane anesthesia, mice were instilled i.n. with 25 μL of PBS containing $1 \times 10^8$ CFUs of COL or COL Δ*seb*. This was followed, 3 days later, by i.n. PR8 inoculation. Separate cohorts of mice were sacrificed after 4 and 10 days to assay for infectious viral titres and IAV-specific $T_{CD8}$ responses, respectively. Animals were monitored for morbidity throughout superinfection experiments.

Finally, in several experiments, mice received $1 \times 10^6$ plaque-forming units (PFUs) of WT VacV in 200 μL of sterile PBS i.p.

## Detection of IAV-specific $T_{CD8}$ by peptide:MHC I multimers

At indicated time points after IAV inoculation, mice were sacrificed by cervical dislocation for their spleen, which was manually homogenized and subjected to treatment with ammonium-chloride-potassium (ACK) lysis buffer to eliminate erythrocytes. Cells were washed, filtered through a strainer with 70-μm pores, and incubated on ice for 20 minutes with a 20-μL aliquot of 2.4G2 hybridoma supernatant to block Fc γ receptors (FcγR) II and III. Alexa Fluor 488-conjugated H-2$K^d$ tetramers loaded with synthetic peptides corresponding to $NP_{147-155}$ or $HA_{518-526}$ (NIH Tetramer Core Facility, Atlanta, GA) were then added to the cells at a 1:200 dilution in PBS containing 5% fetal bovine serum (FBS). $NP_{147-155}$ and $HA_{518-526}$ will be simply referred to as $NP_{147}$ and $HA_{518}$. After 30 minutes on ice, an allophycocyanin-conjugated anti-CD8α mAb (clone 53–6.7) was added to the cells for an additional 30 minutes. In several experiments, cells were co-stained with mAbs to surface CD3ε (clone 145-2C11), CD127 (clone A7R34), Fas (clone 15A7), killer cell lectin-like receptor G1 (KLRG1) (clone 2F1),

lymphocyte activation gene 3 (LAG-3) (clone C9B7W) or programmed cell death-1 (PD-1) (clone J43) or with a mAb detecting intracellular Ki67 (clone SolA15). All fluorochrome-labeled mAbs and isotype controls were purchased from Thermo Fisher Scientific.

In kinetic experiments, tetramer staining was also conducted on serial peripheral blood mononuclear cells (PBMCs) isolated from each animal. Up to 200 μL of peripheral blood was collected from the lateral saphenous vein, immediately mixed with 20 μL of heparin sodium (Sandoz Canada Inc.), diluted 1:2 in PBS, and overlaid on 500 μL of low-endotoxin Ficoll-Paque PLUS (GE Healthcare Life Sciences). Cells were spun at $400 \times g$ for 30 minutes at room temperature, and PBMCs located at the plasma-Ficoll interface were harvested and washed to remove platelets before they were stained.

Where indicated, we employed H-2K$^d$:NP$_{147}$ and H-2K$^d$:HA$_{518}$ dextramers generously provided by Dr. Stephen Haley (Immudex, Copenhagen, Denmark) to detect NP$_{147}$- and HA$_{518}$-specific $T_{CD8}$, respectively. Up to $3 \times 10^6$ splenocytes were incubated first with a 2.4G2 supernatant aliquot and then with 10 μL of dextramers in dark for 10 minutes at room temperature. This step was followed by staining with anti-CD8α mAb (clone 53–6.7) as described above.

Cells were washed to remove excess reagents and immediately interrogated using a BD FACS-Canto II flow cytometer. FlowJo software (Tree Star, Ashland, OR) was used for data analysis.

## Enumeration of interferon (IFN)-γ-producing virus-specific $T_{CD8}$ by intracellular cytokine staining (ICS)

Following immunization or infection with indicated IAVs or VacV, mice were euthanized at indicated time points and erythrocyte-depleted splenocytes or peritoneal exudate cells (PECs) were prepared. In the case of i.n. IAV infection, BAL fluid and lungs were also aseptically collected. BAL was conducted using 1 mL of cold PBS. This procedure was repeated thrice to recover the maximal number of cells. Lungs were then perfused with cold PBS to remove contaminating peripheral blood cells and subsequently homogenized to obtain non-parenchymal cells.

Cells were washed and seeded at $0.3–2.0 \times 10^6$ cells/well of a U-bottom microplate containing RPMI 1640 medium supplemented with 10% heat-inactivated FBS, 2 mM GlutaMAX, 0.1 mM MEM nonessential amino acids, 1 mM sodium pyruvate, 10 mM HEPES, 100 U/mL penicillin and 100 μg/mL streptomycin. Cells were left untreated, exposed to an irrelevant control peptide (NP$_{118-126}$ or simply NP$_{118}$), or stimulated with synthetic peptides corresponding to IAV- or VacV-derived $T_{CD8}$ epitopes (Table 1). NP$_{118}$ is an immunodominant peptide epitope of lymphocytic choriomeningitis virus (LCMV) [48] (Table 1) to which our animals had never been exposed. Each peptide was used at a final concentration of 500 nM. After 2 hours at 37°C, 10 μg/mL of brefeldin A was added to ensure the retention of IFN-γ in the endoplasmic reticulum (ER) of reactivated $T_{CD8}$, and cultures were continued for another 3 hours. Cells were subsequently washed, briefly incubated on ice with 2.4G2 supernatant, and stained for surface CD8α. In several experiments, we additionally used fluorescein isothiocyanate (FITC)-conjugated anti-Vβ mAbs from a Mouse Vβ TCR Screening Panel kit (BD Pharmingen). Next, cells were washed, fixed with 1% paraformaldehyde, permeabilized with 0.1% saponin and stained with an anti-IFN-γ mAb (clone XMG1.2 from Thermo Fisher). The percentage of IFN-γ$^+$ cells was determined after live gating on CD8$^+$ events, which was also used to calculate virus-specific $T_{CD8}$ numbers per organ/sample. Background noise from untreated cells was always negligible and deducted from values obtained from peptide re-stimulation cultures.

## In vivo killing assay

Carboxyfluorescein diacetate succinimidyl ester (CFSE)-based multi-peak *in vivo* killing assays were performed according to a protocol we previously optimized for studying $T_{CD8}$

**Table 1. Peptides used in this study.**

| Virus | Peptide Epitope | Designation | Sequence | Restricting MHC |
|---|---|---|---|---|
| IAV | $NP_{147-155}$ | $NP_{147}$ | TYQRTRALV | H2-K$^d$ |
| IAV | $PB2_{289-297}$ | $PB2_{289}$ | IGGIRMVDI | H2-D$^d$ |
| IAV | $HA_{518-526}$ | $HA_{518}$ | IYSTVASSL | H2-K$^d$ |
| IAV | $NP_{39-47}$ | $NP_{39}$ | FYIQMCTEL | H2-K$^d$ |
| IAV | $HA_{462-470}$ | $HA_{462}$ | LYEKVKSQL | H2-K$^d$ |
| IAV | $NP_{218-226}$ | $NP_{218}$ | AYERMCNIL | H2-K$^d$ |
| VacV | $F2L_{26-34}$ | $F2L_{26}$ | SPYAAGYDL | H2-L$^d$ |
| VacV | $A52R_{75-83}$ | $A52R_{75}$ | KYGRLFNEI | H2-K$^d$ |
| VacV | $E3L_{140-148}$ | $E3L_{140}$ | VGPSNSPTF | H2-D$^d$ |
| LCMV | $NP_{118-126}$ | $NP_{118}$ | RPQASGVYM | H2-L$^d$ |

IAV: influenza A virus; MHC: major histocompatibility complex; LCMV: lymphocytic choriomeningitis virus; VacV: vaccinia virus

immunodominance [26,28,29,49]. Briefly, to prepare control and cognate target cells, erythrocyte-depleted syngeneic splenic cell suspensions were pulsed in separate tubes with 500 nM of $NP_{118}$ or indicated IAV-derived peptides. After 45 minutes at 37°C, cells in each tube were washed, resuspended in PBS and labeled for 15 minutes at 37°C with a specific CFSE concentration. Typically, target cells displaying $NP_{118}$, $HA_{518}$ and $NP_{147}$ were labeled with 0.025, 0.25 and 2 μM of CFSE, respectively. The CFSE reaction was stopped using cold FBS, and target cells were washed and mixed in equal numbers for intravenous (i.v.) tail vein injections into naïve and IAV-primed recipients. Where indicated, target cells were injected into animals that had been given SEB (or PBS), but not PR8, as an additional control. Each mouse received a total of $1 \times 10^7$ cells in 200 μL of PBS. Exactly 1, 2 or 4 hours later, mice were sacrificed for their spleens, which were quickly homogenized and immediately transferred onto ice. Two thousand CFSE$^{low}$ events were acquired on a cytometer, and specific lysis of target cells was determined by the following formula: % specific cytotoxicity = {1 –[(CFSE$^{int/high}$ event number in IAV-primed mouse ÷ CFSE$^{low}$ event number in IAV-primed mouse) ÷ (CFSE$^{int/high}$ event number in naïve mouse ÷ CFSE$^{low}$ event number in naïve mouse)]} × 100.

## IAV-specific IgG2b detection

Three weeks after i.n. infection with PR8, mice were terminally bled via cardiac puncture. Blood specimens were left to clot and then centrifuged at $17,000 \times g$ for 40 minutes at 4°C. IAV-specific IgG2b was detected in serum samples using a previously established protocol [50] and an Invitrogen eBioscience Ready-SET-Go! Kit. MDCK cells were infected with PR8 at a multiplicity of infection (MOI) of 1, harvested 36 hours later, washed and resuspended in PBS containing 1 mM phenylmethylsulfonyl fluoride protease inhibitor (Thermo Fisher), and subjected to three freeze-thaw cycles. The resulting cell lysate was diluted 1:400 in a coating buffer, transferred to wells of an ELISA plate and incubated overnight at 4°C. The wells were subsequently blocked with PBS containing 2% FBS before they were washed using PBS plus 0.05% Tween 20. Serially diluted serum samples were then added for 1 hour at room temperature. After removing unbound serum, a mAb against mouse IgG2b was used to capture bound IgG2b. Next, the plate was washed, and a horseradish peroxidase (HRP)-conjugated anti-mouse IgG polyclonal Ab was added to each well for 3 hours at room temperature. The plate was washed again and received 100 μL/well of a tetramethylbenzidine substrate solution. The enzymatic reaction was stopped using 1 M phosphoric acid, and $OD_{450}$ values were determined using a Synergy H4 Hybrid Microplate Reader.

### Statistical analyses

Statistical comparisons were made using GraphPad Prism 8 software. Student's *t*-tests or one-way Analysis of Variance (ANOVA) with Tukey's post-hoc tests were employed as appropriate. Differences with $p<0.05$ were considered statistically significant.

## Results

### Exposure to SEB before intraperitoneal PR8 immunization amplifies the expansion of 'select' IAV-specific $T_{CD8}$ clones that retain their IFN-γ production and cognate cytolytic capacities

To address how bacterial SAgs alter the magnitude and breadth of primary anti-IAV $T_{CD8}$ responses, we injected cohorts of BALB/c mice i.p. with PBS or with 50 μg of SEB, a prototypic and clinically relevant staphylococcal SAg that is well tolerated in this mouse strain. Three days later, a time point at which SEB-induced T cell proliferation reaches its maximum [51], we immunized mice with PR8, a mouse-adapted IAV strain. We then enumerated cognate $T_{CD8}$ by MHC I multimers or by ICS for IFN-γ at several time points, including on day 7 post-PR8 priming when primary $T_{CD8}$ responses to IAV epitopes peak [29,52] (Fig 1A).

The above model provides multiple advantages. First, unlike in several other laboratory strains, MHC molecules expressed in BALB/c mice have a sufficient binding avidity for SEB [53]. Second, i.p. inoculation of mice with PR8 does not lead to viral propagation, which is due, at least in part, to the modest ability of this virus to infect macrophages and to activate or recruit peritoneal leukocytes [54]. However, it can prime T cells adequately [29,52]. This simulates many seasonal flu vaccination protocols in which inactivated viruses are introduced to the immune system via an unnatural route. Third, the peptide epitopes of PR8 are well-defined in BALB/c mice (Table 1) and give rise to $T_{CD8}$ clones consistently arranged in a reproducible hierarchical order [29,52]. This enables examination of the breadth of IAV-specific $T_{CD8}$ responses. Fourth, since i.p. inoculation of PR8 does not cause active viral infection, cognate $T_{CD8}$ responses could be monitored in the absence of severe complications otherwise ensuing in moribund animals.

We found a single i.p. injection of SEB prior to IAV immunization to raise the frequency of splenic $NP_{147}$-specific $T_{CD8}$ by dextramer staining (Fig 1B). This could be translated, in terms of absolute numbers, to an approximately 3-fold increase over control values (Fig 1C). The observed rise in $NP_{147}$-specific $T_{CD8}$ was also clearly evident in the peripheral blood of SAg-pretreated animals (Fig 1D).

MHC I multimers identify Ag-specific $T_{CD8}$ based solely on their TCR specificity while providing little information about their functional competence. Since SAgs may anergize T cells [17,18], it was important to study T cell responses using functional readouts. Furthermore, although $NP_{147}$ is immunodominant in our system [55], it is not the only epitope recognized in PR8-primed BALB/c mice. Therefore, in the next series of experiments, we tracked functional $T_{CD8}$ recognizing all known peptide epitopes of PR8 by ICS for IFN-γ. Upon incubation with an irrelevant peptide ($NP_{118}$), splenocytes from PR8-inoculated mice did not synthesize any IFN-γ above the background level. In contrast, $NP_{147}$-specific $T_{CD8}$ exhibited a high IFN-γ production capacity (Fig 1E and 1F). Interestingly, SEB administration before PR8 inoculation also 'selectively' increased the frequencies and absolute numbers of $T_{CD8}$ recognizing $HA_{518}$ and $HA_{462}$ without affecting the clonal size of other subdominant $T_{CD8}$ (Fig 1E and 1F). In fact, the observed increase in the case of the $HA_{518}$-specific clone resulted in its 'rank promotion', thus placing it in the second position in the immunodominance hierarchy and only behind $NP_{147}$-specific $T_{CD8}$. We also confirmed the enlarged clonal size of the $HA_{518}$-specific population by dextramer staining (S1A and S1B Fig).

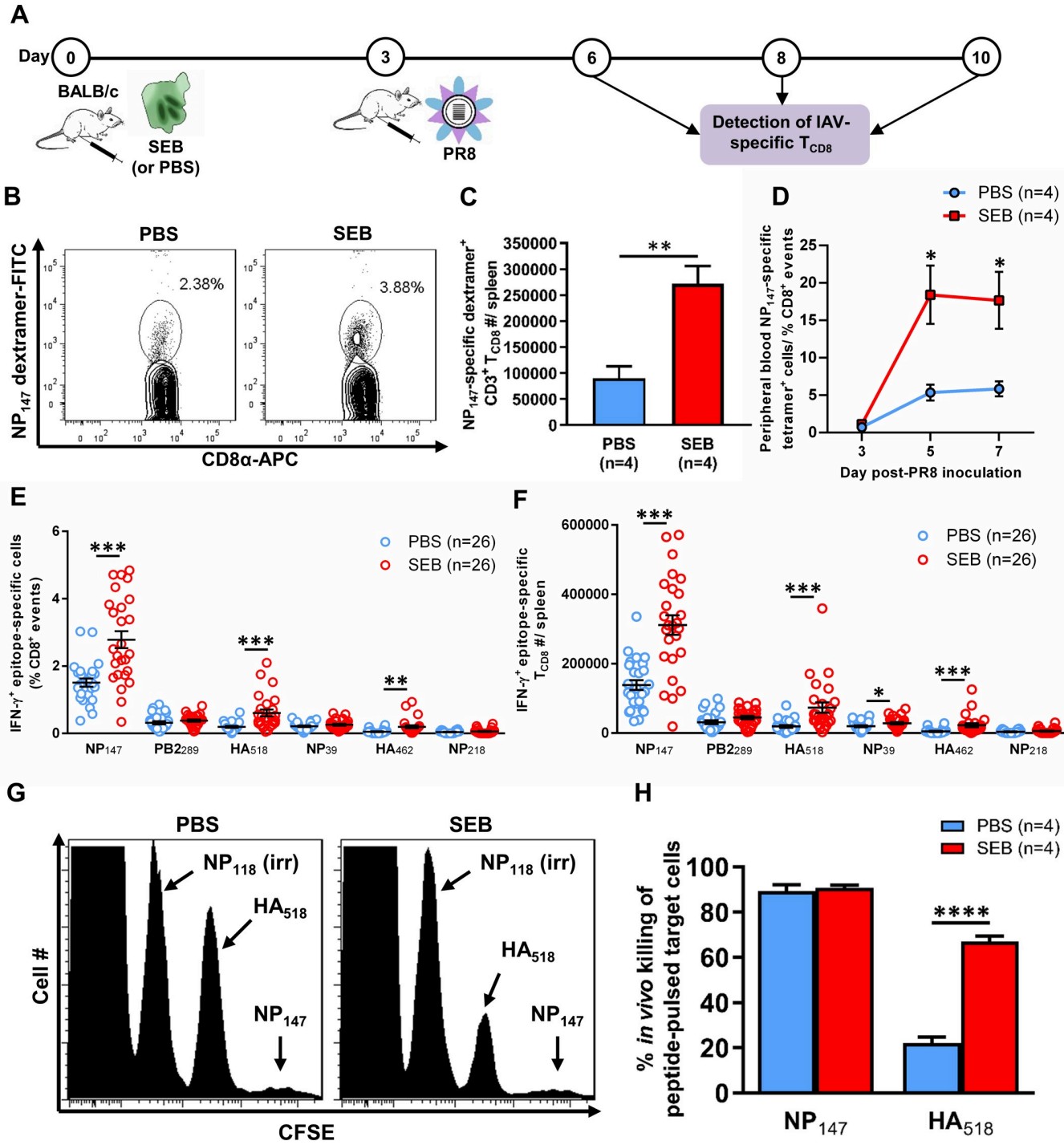

**Fig 1. SEB administration before intraperitoneal PR8 immunization increases the magnitude of primary $T_{CD8}$ responses to select viral epitopes.** (A) BALB/c mice were injected i.p. with PBS or with 50 μg SEB three days before they were immunized i.p. with the PR8 strain of IAV. Seven days later (B,C,E,F,G, H) or at indicated time points (D), splenic (B,C,E,F,G,H) or peripheral blood (D) $T_{CD8}$ responses to indicated PR8 epitopes were evaluated by MHC I dextramer staining (B-C), MHC I tetramer staining (D), intracellular staining for IFN-γ (E-F) or *in vivo* cytotoxicity assays (G-H). Panels B and G illustrate representative contour and histogram plots, respectively. Error bars represent SEM, and *, **, *** and **** denote statistically significant differences with $p<0.05$, $p<0.01$, $p<0.001$ and $p<0.0001$, respectively, using unpaired, two-tailed Student's *t*-tests.

When analyzing IFN-γ production by $NP_{147}$-, $HA_{518}$- and $HA_{462}$-specific $T_{CD8}$ on a per-cell basis, we found comparable geometric mean fluorescence intensity (gMFI) values in PBS- and SEB-pretreated animals. To be exact, the mean gMFI ± SEM of IFN-γ staining in PBS- and SEB-pretreated mice (n = 23 per group) was 3,341 ± 373 *vs.* 3,624 ± 347 ($NP_{147}$), 2,207 ± 277 *vs.* 2,163 ± 298 ($HA_{518}$), and 1,529 ± 210 *vs.* 1,716 ± 240 ($HA_{462}$), respectively. Therefore, SEB had simply expanded the above-indicated IAV-specific $T_{CD8}$ clones without tampering with their ability to produce IFN-γ.

The ultimate function of $T_{CD8}$ is to sense the presence of and destroy target cells displaying cognate peptide:MHC I complexes. We employed a CFSE-based *in vivo* killing assay to test whether SAg-expanded IAV-specific $T_{CD8}$ could fulfill their CTL function. Syngeneic naïve splenocytes were separately pulsed with peptides corresponding to $NP_{147}$, $HA_{518}$ or $NP_{118}$ (irrelevant peptide). Control and cognate target cells thus prepared were washed, mixed in equal ratios and injected i.v. into naïve or PR8-vaccinated mice that had received PBS or SEB earlier according to the schematic timeline depicted in Fig 1A. Four hours later, target cells were tracked based on their differential CFSE intensity. While all three target populations remained intact in naïve mice, $NP_{147}$-coated targets were almost completely absent from the spleens of PR8-primed mice regardless of their prior treatment with PBS or SEB (Fig 1G and 1H). Importantly, SEB pretreatment dramatically augmented $HA_{518}$-specific cytotoxicity (Fig 1G and 1H). As expected, $NP_{118}$-coated target cells were not removed from the spleens. We also confirmed the observed cytotoxicity to be IAV-specific and not due, rather fortuitously, to *in vivo* exposure to a SAg. Following an identical timeline, mice were given SEB or PBS i.p. and left undisturbed for another 10 days in their cages before they were injected i.v. with a mixture of $NP_{118}$-, $NP_{147}$- and $HA_{518}$-pulsed target cells. As anticipated, in the absence of PR8 inoculation, each of the three target cell populations was still present in the spleens of SEB- and PBS-treated animals, almost in equal numbers. To be precise, four hours after the injection of target cells, the mean remaining event number (per corresponding gate) ± SEM in SEB- *vs.* PBS-treated mice (n = 3 per group) was 1,825 ± 35 *vs.* 1,758 ± 37 ($NP_{118}$), 1,524 ± 77 *vs.* 1,383 ± 28 ($NP_{147}$), and 1,434 ± 70 *vs.* 1,419 ± 27 ($HA_{518}$), respectively.

The above results demonstrate that i.p. exposure to SEB expands, rather than deletes, several PR8-specific $T_{CD8}$ clones that exhibit an impeccable IFN-γ production capacity and cytolytic function at the peak of their primary response.

## Systemic SEB administration before intramuscular immunization with heat-inactivated PR8 or intranasal instillation of FluMist enlarges the clonal size of $NP_{147}$-specific/cross-reactive $T_{CD8}$

Although i.p. injection of mice with PR8 is commonly used in murine immunization studies, we sought to extend our work to two additional models of seasonal flu vaccination. First, we administered SEB (or PBS) i.p. to mice that subsequently received an i.m. injection of heat-inactivated PR8 (S2A Fig). Seven days later, a tiny but clearly detectable population of $NP_{147}$-specific $T_{CD8}$ was found in the spleen of SEB-treated mice (S2B and S2C Fig). In a second model, i.p. administration of SEB (or PBS) preceded i.n. instillation of FluMist. Interestingly, once administered to BALB/c mice, the A/California/7/2009 component of this human vaccine yields a PR8-cross-reactive $NP_{147}$-specific response [56]. Ten days after FluMist instillation, $NP_{147}$ tetramer[+] cells were more frequent among $T_{CD8}$ in the lungs of SEB-pretreated mice compared with PBS-injected controls [2.2 ± 0.41% (n = 5) *vs.* 0.63 ± 0.08% (n = 5), *p* = 0.011]. Similarly, more IFN-γ-producing $NP_{147}$-specific $T_{CD8}$ were present in the spleens of SEB-pretreated animals [66,980 ± 10,921 (n = 5) *vs.* 26,070 ± 4,325 (n = 5), *p* = 0.015]. Therefore, pretreatment with SEB amplifies flu-specific $T_{CD8}$ responses in multiple vaccination models.

## SAg-induced clonal expansion of PR8-specific $T_{CD8}$ is dictated by the TCR Vβ families they utilize as opposed to their hierarchical status

The observation that SEB expanded only certain PR8-specific clones (Fig 1E and 1F) suggested to us that this was not a global effect caused by non-specific bystander activation of T cells. We hypothesized that SEB's reactivity with select TCR Vβ families was instead required. To test this hypothesis, we took several approaches.

First, we determined the TCR Vβ usage of the three top-ranked clones in the hierarchy ($NP_{147}$, $PB2_{289}$ and $HA_{518}$). These clones were chosen also because they were impacted differently by SEB (Fig 1E and 1F). Examination of 15 different TCR Vβ segments revealed that $NP_{147}$- and $HA_{518}$-specific $T_{CD8}$, but not $PB2_{289}$-specific cells, preferentially utilized TCR Vβ families that are known to be 'SEB-responsive'. These Vβs are depicted in Fig 2A using various patterns against a red background. The most predominantly utilized Vβs in $NP_{147}$- and $HA_{518}$-specific clones were Vβ8.1/2 and Vβ8.3, respectively. This was apparent in both PBS- and SEB-pretreated animals, but more so in the latter cohort (Fig 2A).

To shed further mechanistic light on this phenomenon, we employed $SEB_{N23A}$ in parallel with SEB in our *in vivo* experiments. $SEB_{N23A}$ is a mutant SAg incapable of binding to mouse TCR Vβ8.2 [11,41]. N23 ($Asn^{23}$) should also mediate the reactivity of SEB with mouse Vβ8.3. This notion is supported by a previous report that several SEB variants carrying point mutations at this position ($SEB_{N23S}$, $SEB_{N23I}$, $SEB_{N23Y}$ and $SEB_{N23K}$) induce only minimal interleukin (IL)-2 production by a Vβ8.3$^+$ hybridoma [40]. To understand how SEB targets and activates mouse T cells expressing Vβ8.3, we superimposed the β-chain from the crystal structure of mouse Vβ8.3/Vα3.3 [57] onto the structure of the homologous mouse Vβ8.2 chain in complex with SEB [58], and then built MHC class II into the complex from the SEB-HLA-DR1 crystal structure [59]. Based on this model (Fig 2B), SEB forms a ternary complex whereby the TCR is 'wedged' apart from the antigenic peptide displayed by MHC II, and the sidechain of SEB N23 engages three residues within CDR2β ($Asn^{54}$, $Leu^{55}$ and $Gln^{56}$), and likely forms an intermolecular hydrogen bond with the side chain of Vβ8.3 $Asn^{54}$. Although these three residues differ between Vβ8.2 and Vβ8.3, critical contacts between SEB and the TCR β-chain are believed to occur through backbone atoms of the β-chain, rather than sidechains, indicating that Vβ recognition by SEB is based mostly on the conformation of the CDR2 loop [60,61]. Therefore, similar to $SEB_{N23S}$, $SEB_{N23I}$, $SEB_{N23Y}$ and $SEB_{N23K}$ [40], our N23 mutant ($SEB_{N23A}$) should be unable to bind Vβ8.3 (Fig 2B). Indeed, SEB, but not $SEB_{N23A}$, enhanced the responsiveness of $NP_{147}$- and $HA_{518}$-specific $T_{CD8}$ (Fig 2C), which are biased towards using Vβ8.1/2 and Vβ8.3, respectively (Fig 2A). Moreover, SEB was significantly more potent than $SEB_{N23A}$ in inducing $HA_{518}$- and $HA_{462}$-specific responses, and a similar trend was observed in the case of $NP_{147}$, which approached but did not reach statistical significance.

In the next series of experiments, we tested two additional SAgs, namely MAM and SEA, in our model. MAM is produced by *Mycoplasma arthritidis*, which is fundamentally different from *S. aureus* in terms of structural and genomic complexity, tissue tropism, and pathogenicity [62]. This SAg offers two additional advantages. First, it has a relatively high affinity for mouse MHC molecules [63]. Second, it binds to Vβ8.1–3 in BALB/c mice [63], the main Vβ families utilized by $NP_{147}$- and $HA_{518}$-specific cells (Fig 2A). These features provided us with a unique opportunity not only to extend our work beyond testing a single SAg (*i.e.*, SEB) but also to solidify our conclusion that SAgs expand IAV-specific $T_{CD8}$ in a TCR Vβ-specific fashion. As hypothesized, pre-exposure to MAM selectively enhanced $NP_{147}$- and $HA_{518}$-specific responses (Fig 2D), which mirrored our findings in SEB-pretreated mice (Fig 1F).

SEA, on the other hand, binds to Vβ1, 3, 10, 11 and 17 [64]. Of these, Vβ3 and Vβ17 are responsive to both SEA and SEB. However, they are scarce or absent in BALB/c mice, due to

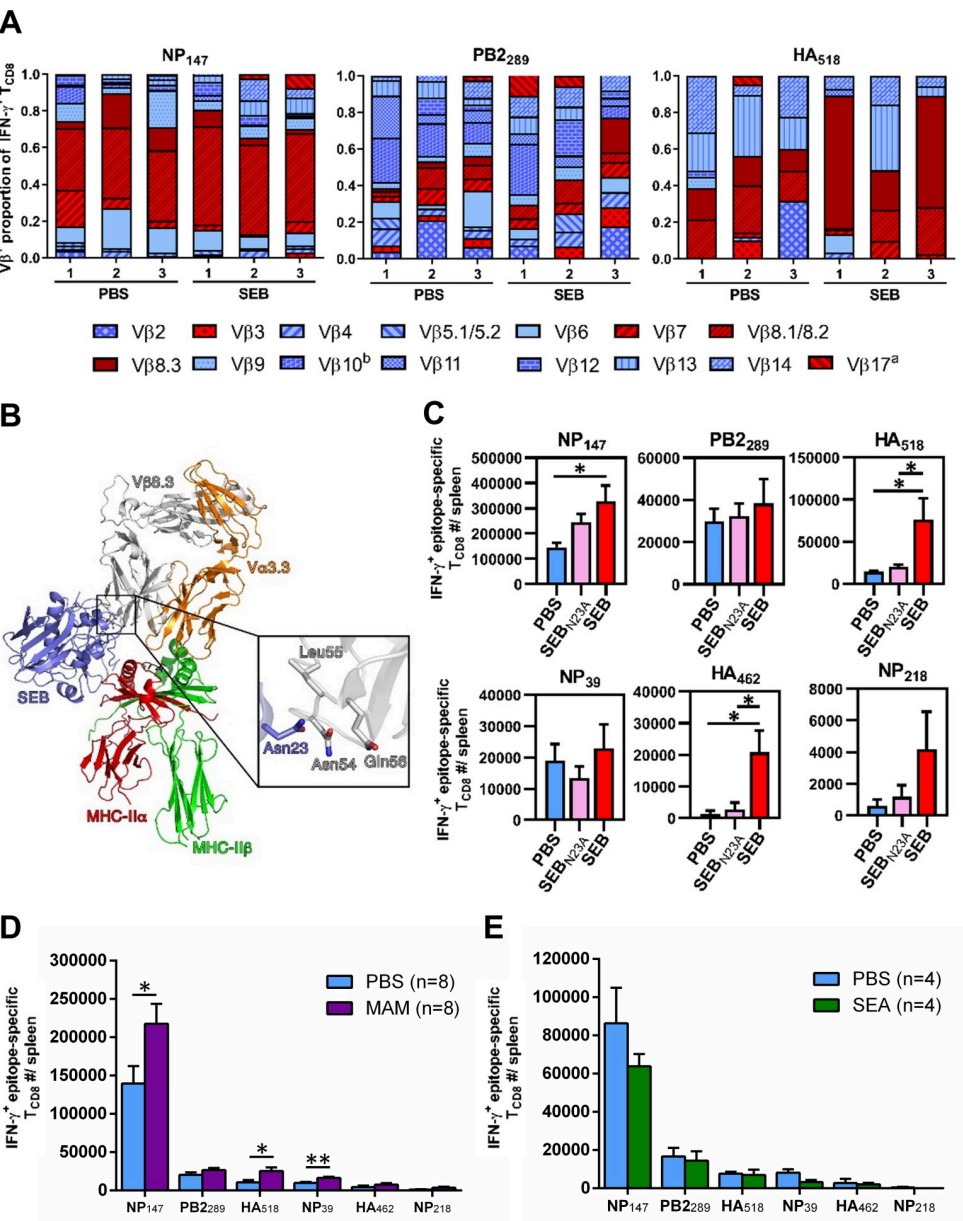

**Fig 2. Preferential TCR Vβ usage by T_CD8 clones dictates their enhanced or unaltered responsiveness to PR8 following exposure to bacterial SAgs.** (A) Mice were injected i.p. with PBS or with SEB. Seven days later, NP_147-, PB2_289- and HA_518-specific T_CD8 were identified by ICS for IFN-γ and also surface-stained with 15 mAbs recognizing indicated TCR Vβ regions. The abundance of each Vβ region is expressed as a proportion of all detectable Vβs for each epitope-specific T_CD8 population in individual mice (n = 3 per group). SEB-binding Vβs are shown using different patterns against a red background. (B) A ribbon diagram model of SEB (blue) in complex with mouse TCR Vβ8.3 (gray) and the MHC class II molecule HLA-DR1 (α-chain, red; β-chain, green) was constructed. The cognate peptide displayed within the open groove of HLA-DR1 is shown in black. The insert illustrates the key position of Asn[23] for interactions with indicated residues of the CDR2β region of mouse Vβ8.3. (C) Parallel cohorts of mice (n = 5 per group from two independent experiments) were injected i.p. with PBS, 50 μg of SEB, or 50 μg of SEB_N23A, followed 3 days later by PR8 immunization i.p. After 7 days, T_CD8 responses to indicated epitopes were quantified by ICS for IFN-γ. (D) Mice were injected i.p. with PBS or with 50 ng of MAM before they received, 3 days later, an i.p. inoculum of PR8. On day 7 post-PR8 immunization, epitope-specific T_CD8 were enumerated. Data pooled from two independent experiments are shown. (E) Three days before PR8 inoculation, mice were injected with PBS or with 50 μg of SEA i.p. Seven days after PR8 immunization, epitope-specific T_CD8 were enumerated. Error bars represent SEM. * and ** denote significant differences with $p < 0.05$ and $p < 0.01$, respectively, using unpaired, two-tailed Student's t-tests (D) or one-way ANOVA followed by Tukey's post-hoc tests (C).

the presence of certain endogenous retroviruses in this strain [65,66]. Since the remaining SEA-reactive Vβ families are not expressed by $NP_{147}$- and $HA_{518}$-specific $T_{CD8}$ (Fig 2A), SEA should not alter the response magnitude of these clones, which is exactly what we observed (Fig 2E).

Taken together, the above results demonstrate that SAg-triggered expansion of IAV-specific $T_{CD8}$ is mechanistically linked to their TCR Vβ makeup.

## SEB invigorates select epitope-specific $T_{CD8}$ responses to WT and reassortant IAV strains

Our experiments thus far involved only PR8, an H1N1 strain suitable for mouse studies and for propagation in chicken eggs. It was therefore pertinent to explore whether pre-exposure to SEB could similarly enhance $T_{CD8}$ responses to other IAV strains, including H3N2 viruses that are included in seasonal flu vaccine formulations. To this end, we tested several WT and recombinant viruses. These included WT NT60 and HK strains whose HA and neuraminidase (NA) together comprise an H3N2 coat that is antigenically dissimilar to the H1N1 coat of PR8 (Fig 3A). Another cohort of mice was vaccinated with X31, a reassortant virus that contains a PR8 core and an HK coat [35]. Finally, we used the reassortant H3N1 strain J-1, which contains seven PR8 genes along with an HK-derived HA [36]. As illustrated in Fig 3B–3E, a sizeable response to $NP_{147}$ was evident following immunization with NT60, HK, X31 or J-1, which all expressed a nucleoprotein (NP) containing this immunodominant peptide. Importantly, pretreatment with SEB enhanced this response in all cases (Fig 3B–3E). In contrast, $NP_{39}$- and $NP_{218}$-specific responses were not augmented. This was consistent with the findings from our PR8 vaccination experiments (Fig 1E and 1F). Finally, none of the H1-derived epitopes elicited a measurable response in PBS- or SEB-treated mice (Fig 3B–3E). This was expected since NT60, HK, X31 and J-1 strains all express H3, not H1 (Fig 3A). The above results demonstrate that cognate $T_{CD8}$ responses to multiple IAV strains can be intensified by SEB.

## SAgs can augment or attenuate recall anti-IAV $T_{CD8}$ responses depending on when they are encountered

Infection with or vaccination against one IAV serotype may induce cross-protection against another, which is commonly referred to as heterosubtypic immunity.

In our primary vaccination model, introducing SEB three days before PR8 led to augmented responses against $NP_{147}$ and $HA_{518}$ (Fig 1A–1H). Based on our kinetic analyses, $NP_{147}$- and $HA_{518}$-specific clones remained significantly enlarged in the spleen of SEB-pre-treated animals after day 7 (Fig 4A and 4B). Therefore, we asked whether pre-exposure to SEB affects the progression of IAV-specific $T_{CD8}$ to a memory state. We found that at the peak of their primary response, both $NP_{147}$- and $HA_{518}$-specific $T_{CD8}$ pools contained more $CD127^{high}KLRG1^{low}$ cells and less $CD127^{low}KLRG1^{high}$ cells in SEB-treated mice (Fig 4C). These populations are considered memory precursors and short-lived terminal effectors, respectively, in antiviral defense [67].

The above findings prompted us to hypothesize that SEB administration before IAV inoculation should increase the precursor frequency of memory $T_{CD8}$. We used a prime-boost strategy to test this hypothesis. Accordingly, mice were injected with SEB, primed with PR8 and boosted 30 days later with SEQ12 (Fig 4D). SEQ12 was initially generated through sequential passages of PR8 in the presence of anti-HA mAbs [34]. Therefore, it cannot be efficiently neutralized by anti-HA Abs produced by mice during the priming phase. This prevents the removal of the virions before they can induce an anamnestic $T_{CD8}$ response. It is also noteworthy that SEQ12 has accumulated mutations within its HA's globular domain while retaining

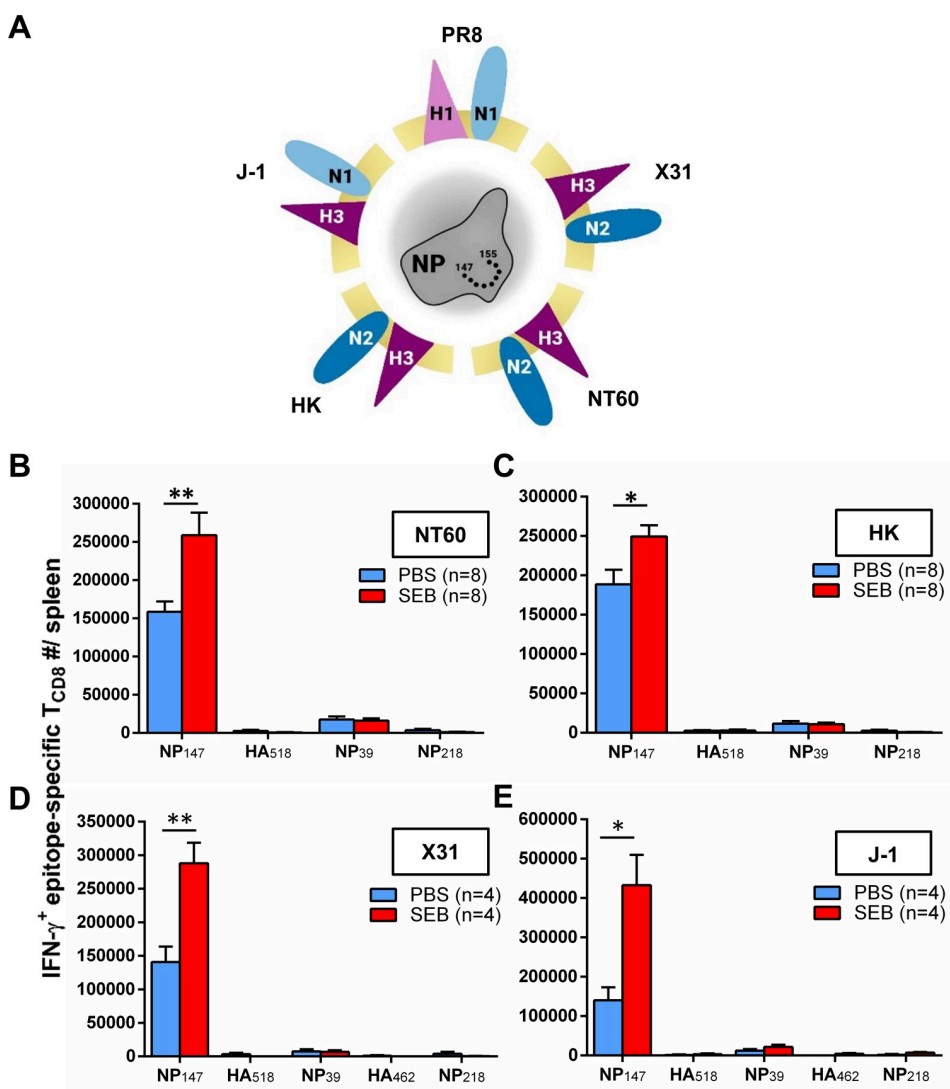

**Fig 3. Systemic pre-exposure to SEB augments the primary NP$_{147}$-specific response following immunization with multiple IAV strains.** Mice were injected i.p. with PBS or SEB followed, 3 days later, by i.p. immunization with WT or recombinant IAV strains expressing the same NP (A). These strains were NT60 (B), HK (C), X31 (D) and J-1 (E). Seven days after IAV inoculation, the absolute numbers of IFN-γ-producing epitope-specific T$_{CD8}$ were calculated. Error bars represent SEM. * and ** denote significant differences with $p < 0.05$ and $p < 0.01$, respectively, using unpaired Student's $t$-tests.

the structure of this membrane glycoprotein and identical sequences in and around HA$_{518}$, one of our epitopes of interest. Therefore, the response to HA$_{518}$ should be preserved as we previously reported [29].

As hypothesized, heterologous prime-boost vaccination with PR8 and SEQ12 resulted in a more rigorous secondary response to both NP$_{147}$ and HA$_{518}$ compared with the primary response to PR8 (Fig 4E and 4F and Fig 1E and 1F). Furthermore, seven days after SEQ12 inoculation, SEB-pretreated mice had significantly higher proportions and absolute numbers of NP$_{147}$- and HA$_{518}$-specific cells in their splenic T cell pools (Fig 4E and 4F). Together, the above results demonstrate that when a SAg is introduced before a primary response to an IAV strain is mounted, subsequent responses to unrelated IAV serotypes harboring identical core

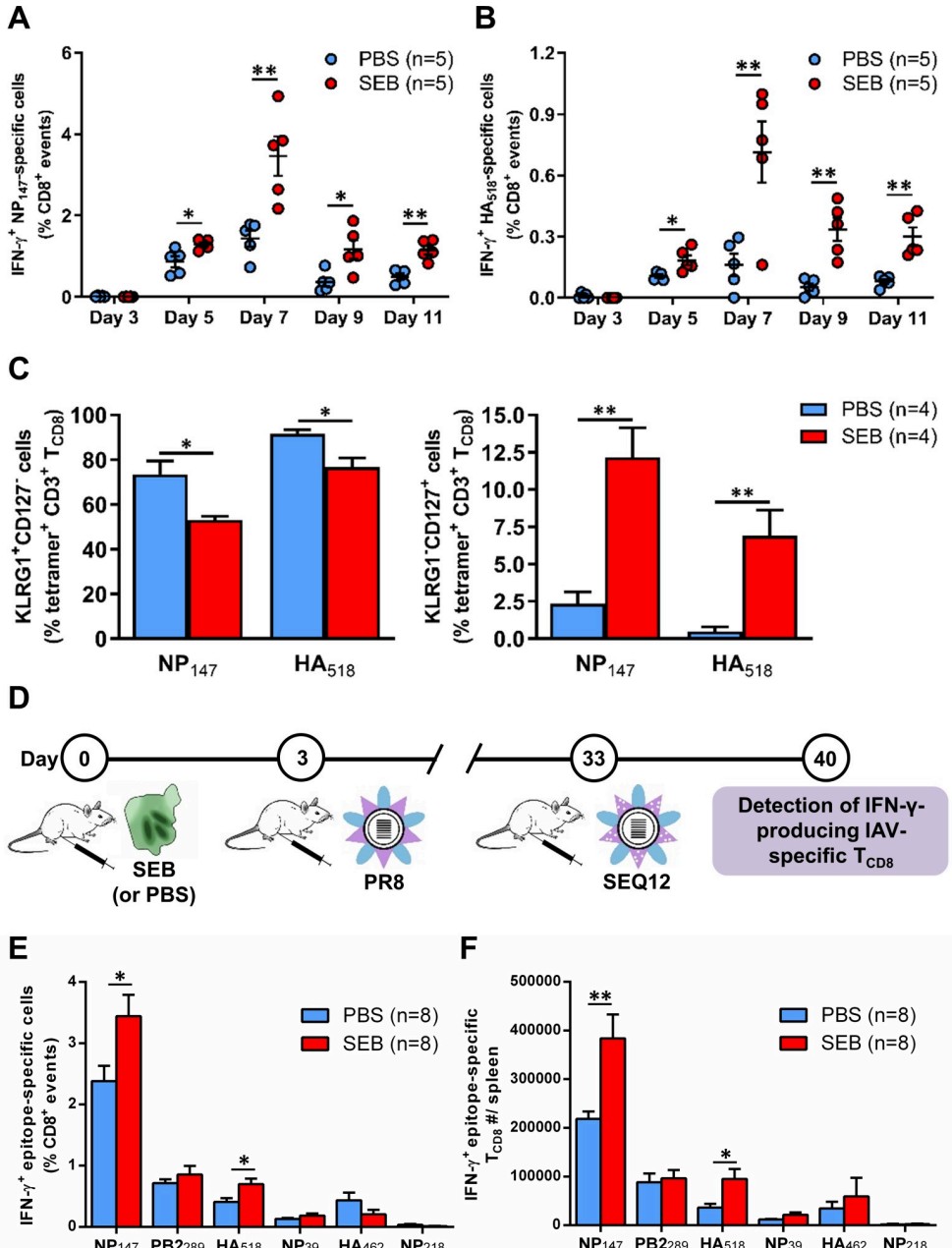

**Fig 4. Pre-exposure to SEB raises the frequency of memory T_CD8 precursors and invigorates recall responses of SEB-binding IAV-specific T_CD8.** (A-B) Mice were injected i.p. with PBS or with 50 μg of SEB before they received i.p. immunization with PR8. On indicated days, members of each cohort were sacrificed for their spleens in which the frequencies of NP_147-specific (A) and HA_518-specific (B) T_CD8 were determined by ICS for IFN-γ. Each symbol represents an individual mouse, and data are pooled from two independent experiments yielding similar results. (C) On day 7 post-PR8 immunization, the frequencies of KLRG1+CD127- T_CD8 (short-lived effectors) and KLRG1-CD127+ T_CD8 (memory precursors) were determined after surface staining of splenic cells with mAbs to CD3 and CD8α along with MHC I tetramers detecting NP_147- or HA_518-specific cells. (D) Mice were injected with PBS or SEB three days before they were primed with PR8 i.p. Thirty days later, boosting immunization with SEQ12 was conducted using the same injection route. On day 7 post-challenge with SEQ12, the frequencies (E) and absolute numbers (F) of IFN-γ-producing epitope-specific T_CD8 in the spleens were determined. The results depicted in E-F are pooled from two independent experiments. Error bars (A-C and E-F) represent SEM. * and ** denote significant differences with $p<0.05$ and $p<0.01$, respectively, using unpaired Student's $t$-tests.

protein epitopes can be enhanced, provided that $T_{CD8}$ specific for such epitopes utilize SAg-responsive TCR Vβ families.

Next, we explored the effect of SEB on recall responses when this SAg is 'seen' before a secondary anti-IAV response gets underway. Mice were primed with PR8, given SEB, and then boosted with SEQ12 (Fig 5A). Remarkably, seven days after boosting, the frequencies and absolute numbers of IAV-specific IFN-γ-producing $T_{CD8}$ were drastically reduced, a finding that was manifest across all tested epitopes regardless of their hierarchical position or TCR Vβ usage preferences (Fig 5B and 5C). To rule out the possibility that injection with SEB shortly (3 days) before SEQ12 may unusually weaken the responsiveness of $T_{CD8}$ to this particular strain irrespective of their naïve versus memory status, we examined the primary response to SEQ12. In this control experiment, naïve mice received SEB or PBS three days before they were inoculated with SEQ12 (S3A–S3C Fig). Similar to primary immunizations with PR8, NT60, HK, X31 and J-1 (Fig 1 and Fig 2), select cognate responses to SEQ12 were augmented. Therefore, the results shown in Fig 5B and 5C reflect a unique feature of anamnestic $T_{CD8}$ responses launched shortly after exposure to SEB.

To dissect this hyporesponsive phenotype, we examined the expression levels of several proliferation, death and exhaustion markers among detectable $T_{CD8}$. Cognate tetramer$^+$ $T_{CD8}$ from SEB-treated animals showed diminished expression of Ki67 but elevated levels of Fas and LAG-3 on a per-cell basis (Fig 5D). In addition, there was a trend towards increased PD-1, especially in $HA_{518}$-specific cells. Therefore, exposure to SEB may impede the proliferative capacity of anti-IAV memory $T_{CD8}$ and render them prone to death or exhaustion when animals encounter another IAV serotype with conserved internal epitopes.

The above findings indicate that naïve and memory $T_{CD8}$ are qualitatively different in their susceptibility to a 'double hit' with a SAg and IAV within a short time span.

## Exposure to SEB after IAV inoculation raises the frequency of functional antiviral $T_{CD8}$

In certain scenarios, exposure to SAgs may succeed a recent encounter with a new IAV strain, due for instance to vaccination. To determine how $T_{CD8}$ react to IAV in this case, we inoculated mice with PR8 followed, three days later, by an SEB injection. Seven days later, PR8-specific $T_{CD8}$ were enumerated by ICS, and their *in vivo* cytolytic function was also assessed (Fig 6A). We found a dramatic increase in both the frequencies and the absolute numbers of IFN-γ-producing $T_{CD8}$ recognizing multiple PR8 epitopes (Fig 6B and 6C). Furthermore, $NP_{147}$-displaying target cells were much more readily eliminated in SEB-treated mice (Fig 6D and 6E). This was evident at several time points after cognate target cells were injected into PR8-primed animals (Fig 6E). Therefore, functionally competent IAV-specific CTL pools are expanded following an *in vivo* exposure to SEB.

## Local but not systemic exposure to SEB augments pulmonary $T_{CD8}$ responses to active IAV infection

We next sought to determine whether exposure to SEB affects $T_{CD8}$ responses to IAV infection as opposed to vaccination. Separate cohorts of mice were left uninfected or injected i.p. with PBS or SEB before they received a sublethal i.n. inoculum of PR8. As anticipated, while uninfected mice showed a slight weight gain in the course of these experiments, infection resulted in a gradual weight loss, which was worst at around day 7 post-PR8 inoculation and which was partially reversed afterwards (Fig 7A). A control cohort receiving only SEB exhibited a minimal and insignificant weight loss. In addition, the extent and kinetics of weight loss and recovery were comparable between infected mice that had received PBS or SEB (Fig 7A). Finally,

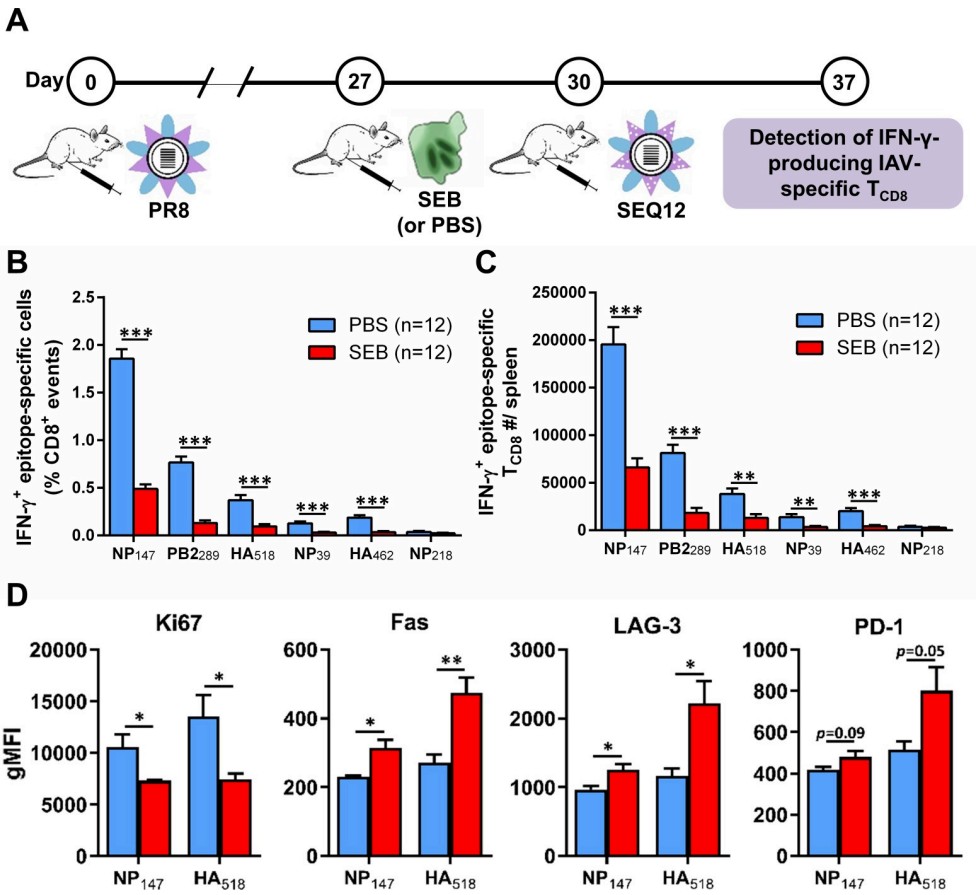

**Fig 5. Exposure to SEB before boosting immunization upregulates Fas and exhaustion markers on anti-IAV $T_{CD8}$ and compromises their anamnestic responses.** (A) Mice were primed with PR8, injected with either PBS or 50 µg of SEB, and then boosted with SEQ12 through the i.p. route according to the schematically depicted schedule. Seven days after SEQ12 inoculation, animals were sacrificed for their spleens in which the frequencies (B) and the absolute numbers (C) of IFN-γ-producing epitope-specific $T_{CD8}$ were determined by ICS. (D) $NP_{147}$- and $HA_{518}$-specific $T_{CD8}$, which were identified by MHC I tetramers, were co-stained with mAbs against intracellular Ki67 and surface Fas, LAG-3 and PD-1. Error bars represent SEM (n = 4 per group). *, ** and *** denote statistically significant differences with $p < 0.05$, $p < 0.01$ and $p < 0.001$, respectively, using unpaired Student's $t$-tests.

systemic pre-exposure to SEB did not alter the viral titres within the lungs and BAL of PR8-infected animals (Fig 7B).

On day 10 post-PR8 infection, which corresponds to the peak of the anti-IAV response in the lungs [45], IFN-γ-producing epitope-specific $T_{CD8}$ numbers were nearly equal in the lungs and within the BAL fluid of PBS- and SEB-treated mice (Fig 7C). Interestingly however, SEB administration almost doubled splenic $NP_{147}$-specific T cell numbers (Fig 7C). In this setting, only a trend towards an increased response to $HA_{518}$ was noticeable in SEB-primed animals. These data indicate that i.p. pretreatment with SEB can enhance the systemic, but not the local, immunodominant $T_{CD8}$ response in the aftermath of a respiratory IAV infection.

Intraperitoneally injected SEB can reportedly reach the lungs and increase vascular permeability and leukocyte accumulation in mice [68]. However, it failed to alter PR8-specific $T_{CD8}$ numbers in the lungs or in BAL in our model as indicated above, which was curious. We posited that pulmonary IAV-specific $T_{CD8}$ responses may be influenced more forcibly by local SEB administration. Injection of SEB via the i.n. route has been used in the past to model exposure to this SAg due to bacterial colonization or through inhalation of aerosolized SEB

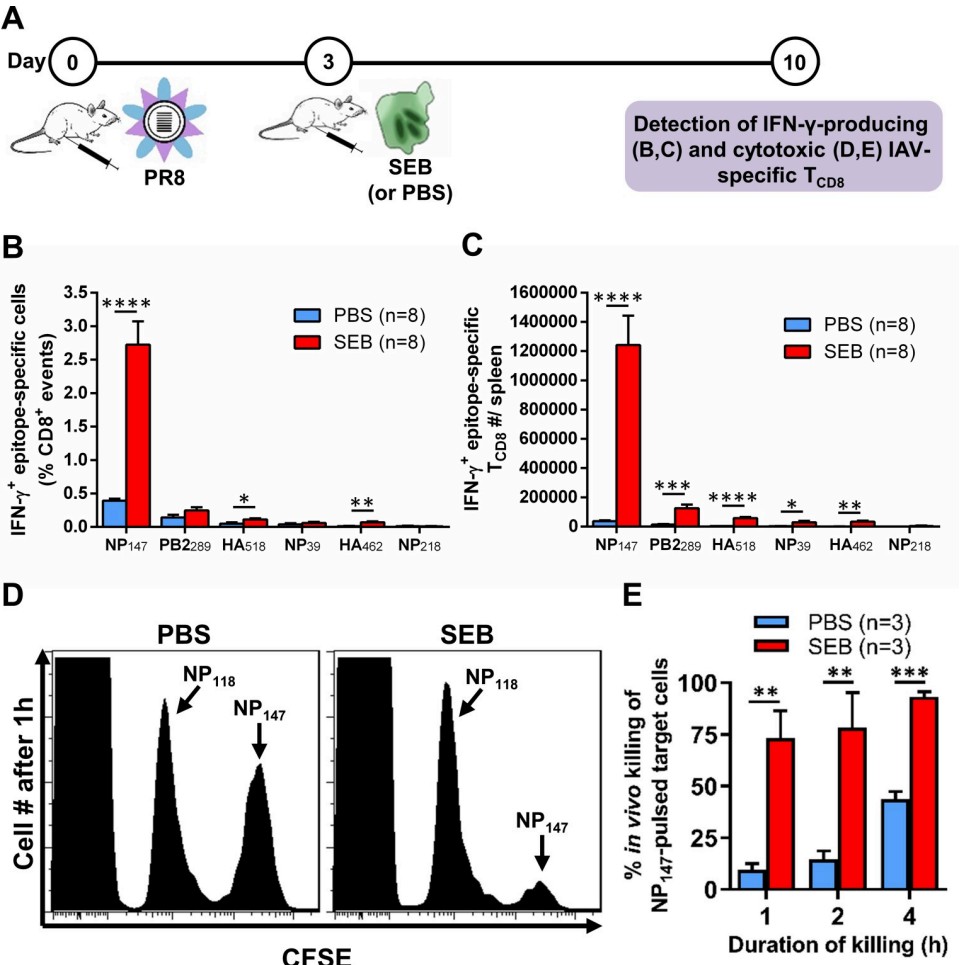

**Fig 6. Exposure to SEB after PR8 immunization augments IAV-specific $T_{CD8}$ responses.** (A) Mice were inoculated with PR8 and then injected with PBS or SEB (50 μg) i.p. Seven days after the SEB challenge (or after PBS injection), animals were euthanized, and PR8-specific $T_{CD8}$ were enumerated by ICS for IFN-γ in each spleen. Data pooled from two independent experiments yielding similar results are shown in B-C. In separate experiments, CFSE$^{low}$ naïve splenocytes displaying $NP_{118}$ (control target cells) and CFSE$^{high}$ naïve splenocytes pulsed with $NP_{147}$ (cognate target cells) were mixed in equal numbers and injected i.v. into PR8-inoculated mice that had subsequently received PBS or SEB. Mice were sacrificed after 1, 2 or 4 hours for their spleens in which target cells were identified based on their differential CFSE intensities. (D) Representative histograms from a 1-hour *in vivo* killing assay are depicted. (E) *In vivo* cytotoxicity at indicated time points is averaged and shown for 3 mice/group/time point. Error bars represent SEM. *, **, *** and **** denote differences with $p<0.05$, $p<0.01$, $p<0.001$ and $p<0.0001$, respectively, using unpaired Student's *t*-tests.

maliciously employed as a bioweapon [69,70]. In fact, SEB is regarded as a 'category B priority bioterrorism agent' by the Centers for Disease Control and Prevention [71]. In our system, i.n. injection of SEB did not change the weight loss pattern of PR8-infected mice (Fig 8A and 8B) but now increased $NP_{147}$-specific $T_{CD8}$ numbers in the lungs consistent with an augmented local response (Fig 8C).

Finally, we examined how SEB may affect anti-PR8 T cell responses in the context of bacterial colonization/superinfection. To do so, we injected B6 mice i.n. with either WT *S. aureus* COL or an SEB deletion mutant of this bacterium (COL *Δseb*) that we previously generated [37]. These strains exhibited identical growth rates when cultured *in vitro* (Fig 9A). We found that the lungs and BAL of mice that were infected i.n. with PR8 following infection with either

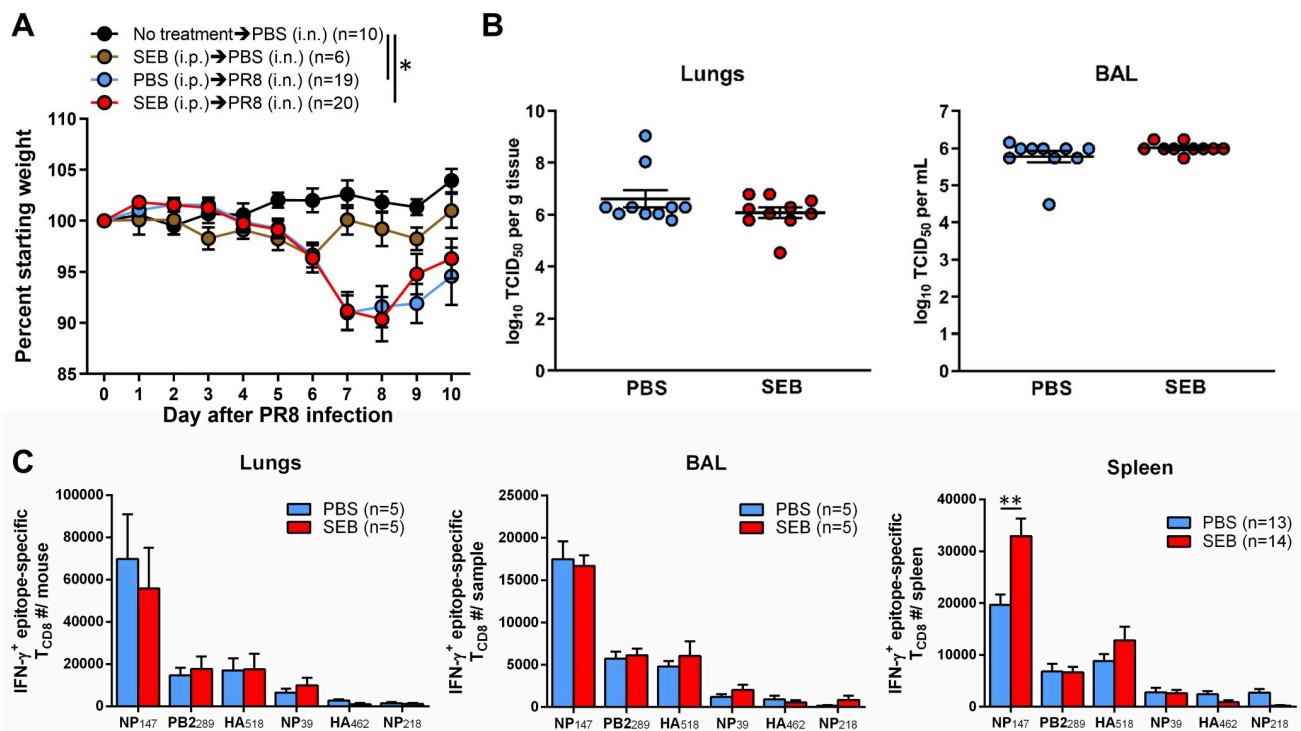

**Fig 7. Intraperitoneal administration of SEB before sublethal PR8 infection enhances the splenic, but not the pulmonary, NP_{147}-specific response.**
(A) Indicated numbers of BALB/c mice received no treatment or injected with PBS or SEB i.p. followed, 3 days later, by intranasal instillation of PBS or a 25-μL inoculum of infectious allantoic fluid approximating 0.3 MLD_{50} of PR8. Weight loss was monitored for 10 days. (B) On day 4 post-infection, viral titres in the lungs and BAL were determined using a modified TCID_{50} assay detailed in Materials and Methods. Each symbol represents an individual mouse. (C) Ten days after infection, lungs, spleens and BAL fluid were collected from separate cohorts from 5 independent experiments. Epitope-specific T_CD8 were enumerated in each spleen and also in pooled lung and BAL samples by ICS for IFN-γ. Error bars represent SEM. * and ** denote significant differences with $p<0.05$ and $p<0.01$, respectively, using one-way ANOVA (A) or an unpaired Student's $t$-test (C).

COL or COL *Δseb* contained comparable infectious viral titres (Fig 9B). Nevertheless, our tetramer staining and ICS assays revealed either a statistically significant decline or a trend towards a decrease in pulmonary and splenic NP_{147}-specific T_CD8 in COL *Δseb*-infected mice (Fig 9C).

All together, the above results indicate that pulmonary PR8-specific T_CD8 responses can be impacted by SEB if this SAg is introduced locally as a stand-alone toxin or after it is released in the airway by colonizing or infectious *S. aureus* bacteria.

## Discussion

How bacterial SAgs modify bulk or Ag-specific T cell responses has been a subject of several investigations in the past. Anergy, TCR downregulation and apoptotic death have often been reported as outcomes of exposure to these potent exotoxins, which may contribute to either a transient state of unresponsiveness or longer-term immunosuppression [12,16–18,72,73]. Paradoxically, Ag-specific T cell growth and activation have also been reported in the face of a challenge with SAgs. For instance, Coppola and Blackman reported that splenic CD8+CD62L^{lo}CD44^{hi} bulk memory cells from X31-infected mice kill IAV-infected cells more efficiently when they are exposed to SEB *in vitro* [74]. We recently demonstrated that 'preexisting' antiviral memory T_CD8 can be expanded and activated by bacterial SAgs in mice and among human PBMCs in the absence of a recall challenge [19]. How SAgs affect the magnitude and breadth of vaccine- and infection-elicited *in vivo* recall responses, which mediated

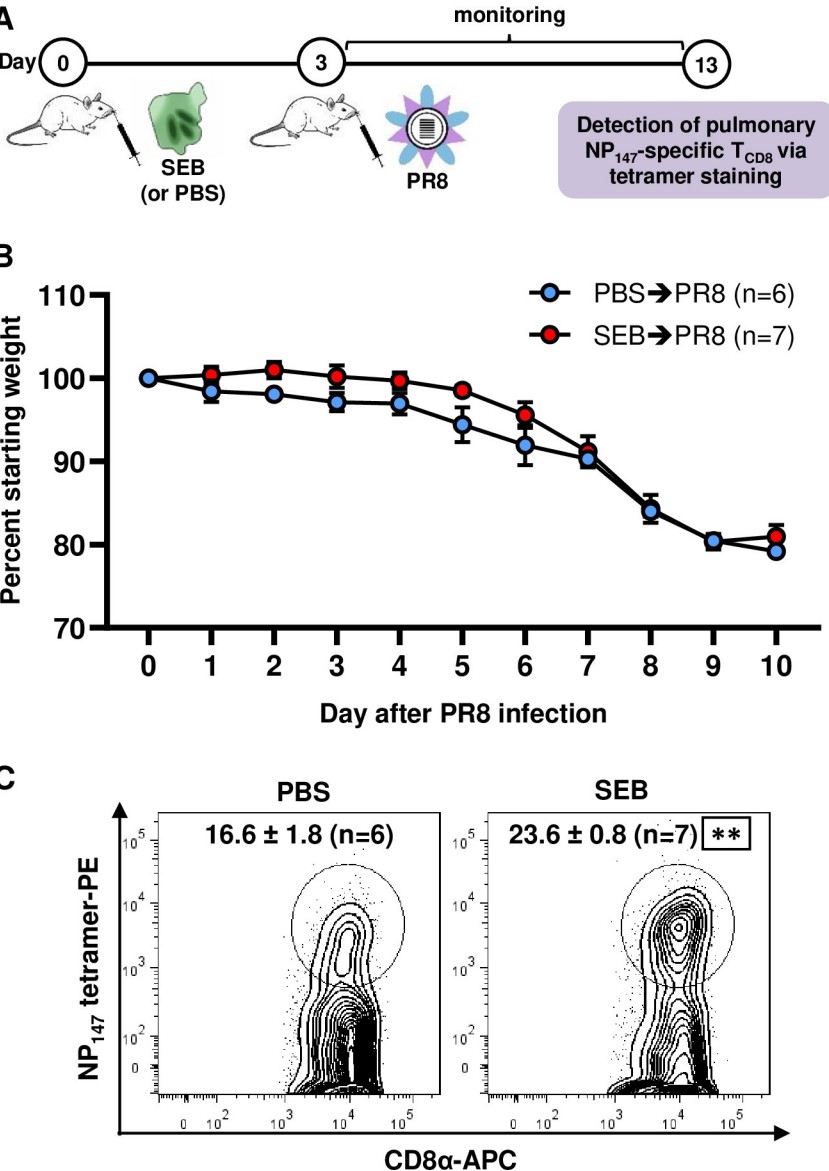

**Fig 8. Intranasal instillation of SEB before PR8 does not worsen infection-induced weight loss but increases the pulmonary response to $NP_{147}$.** (A) BALB/c mice were injected i.n. with 50 ng of SEB in 25 μL PBS (n = 7) or with PBS alone (n = 6) three days before they were sublethally infected with PR8. Animals were monitored daily after PR8 infection, and their weight loss was recorded (B). On day 10 post-infection, the percentages of $NP_{147}$-specific $T_{CD8}$ were determined in the lungs of PBS- and SEB-pretreated mice by MHC I tetramer staining (C). Representative contour plots are depicted, and mean ± SEM values are indicated. ** denotes a significant difference between PBS- and SEB-treated mice with $p<0.01$ using an unpaired Student's $t$-test.

heterosubtypic immunity and potentially cross-protection, has been unclear and a subject of our current work.

Another important question is whether SAgs may manipulate or reshape the immunodominance hierarchies of Ag-specific $T_{CD8}$, including IAV-specific cells, which in turn determines the narrowness or breadth of primary and recall anti-IAV responses. Although immunodominance has been often investigated in experimental settings using mouse models, increasing evidence suggests that it is also relevant to human anti-IAV responses [75,76,77].

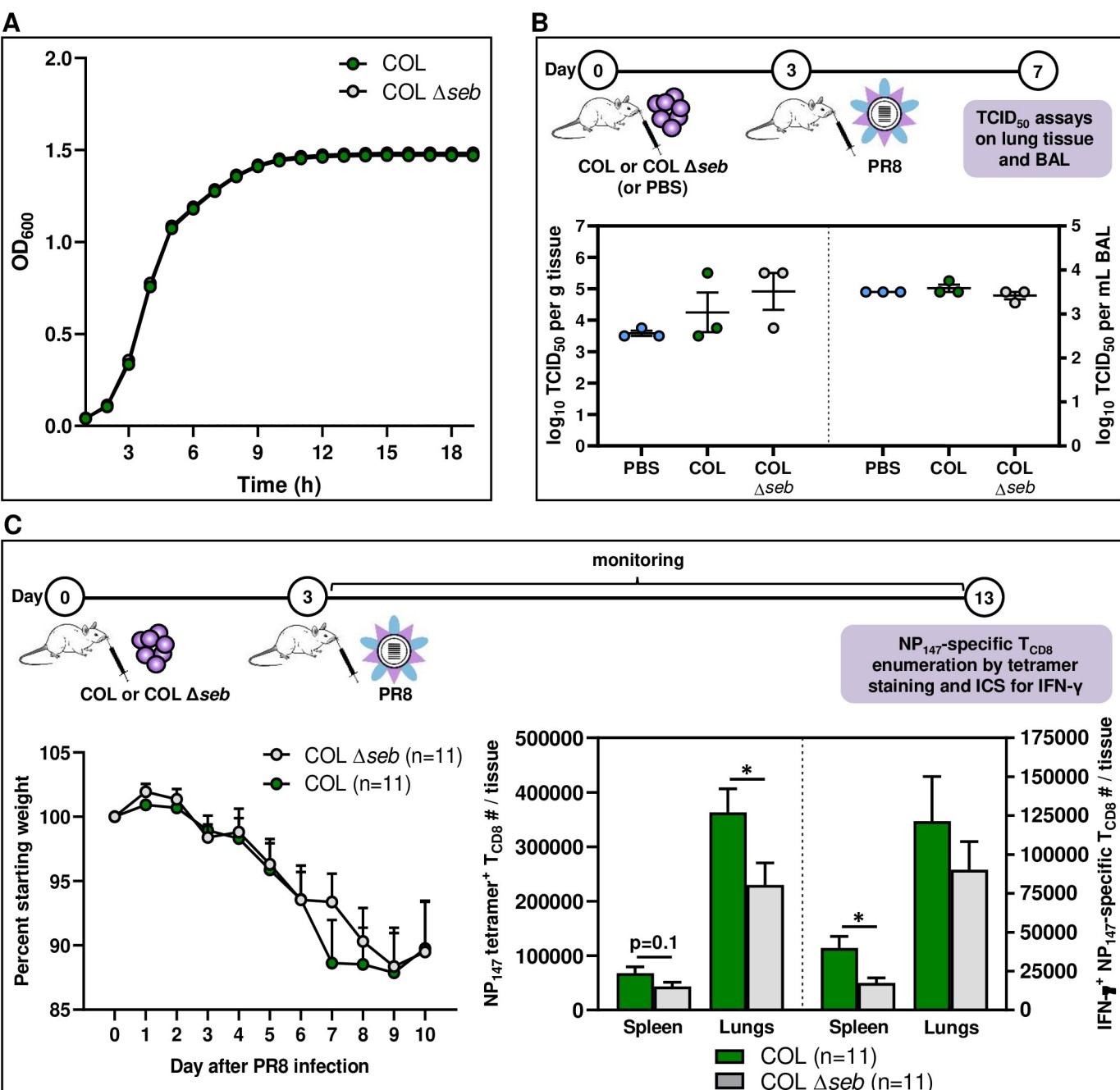

**Fig 9. Respiratory infection with an SEB deletion mutant of *S. aureus* before PR8 results in weaker NP_{147}-specific T_{CD8} responses.** (A) COL and COL Δ*seb* were cultured in half-BHI medium, and their growth rates at 37°C were monitored. OD_{600} readings were used to generate a growth curve for each strain. Mice were then given PBS, $1 \times 10^8$ CFUs of COL or $1 \times 10^8$ CFUs of COL Δ*seb* i.n. three days before they were infected sublethally with PR8. On day 4 post-PR8 infection, mice (n = 3/group) were sacrificed for their lungs and BAL fluid in which infectious PR8 titres were determined by TCID_{50} assays. Each symbol represents an individual mouse (B). Separate cohorts of mice were monitored for weight loss until day 10 post-PR8 infection (C), at which point animals were sacrificed for their spleens and lungs in which NP_{147}-specific T_{CD8} were enumerated by MHC I tetramer staining or ICS for IFN-γ (C). Mean ± SEM values are depicted for 11 mice per group pooled from three independent experiments yielding similar results. Statistical comparisons between COL- and COL Δ*seb*-infected mice were carried out using an unpaired Student's *t*-test, and * denotes a significant difference with $p < 0.05$.

Of note, IAVs are highly mutating viruses whose HA and NA constantly change, in a process called antigenic drift, to avoid detection by neutralizing Abs. In contrast, internal proteins,

such as NP, are highly conserved. However, this does not necessarily mean that they never mutate. In fact, mutations in HLA-B27- and HLA-B35-restricted NP-derived peptides to facilitate escape from CTL recognition have been reported [78,79]. In this investigation, we found pre-exposure to SEB and MAM to selectively enhance primary responses to $NP_{147}$ and $HA_{518}$, which promoted the hierarchical rank of the latter. Importantly, which $T_{CD8}$ clones could be expanded by the above SAgs was mechanistically linked to the TCR Vβ composition of $T_{CD8}$ as opposed either to their default hierarchical status or to the propensity of the protein source of the epitope they detect, namely NP and HA. It will be interesting to assess whether SAg-expanded IAV-specific $T_{CD8}$ will be equally likely, less likely or more likely to undergo mutations, which may potentially affect their significance in the context of influenza following exposure to the toxins or bacterial pathogens that release them.

To investigate how SAgs influence $T_{CD8}$ responses in heterosubtypic immunity to IAV, we administered SEB at two different but critical time points, which yielded opposite results. Exposure to SEB before priming augmented recall responses to $NP_{147}$ and $HA_{518}$. This was consistent with increased frequencies of $T_{CD8}$ responding to these epitopes at and after their primary response peak and with a differentiation program favoring the generation of $KLRG1^-CD127^+$ memory precursors. On the contrary, SEB administration prior to boosting immunization drastically diminished IAV-specific $T_{CD8}$ responses, which was accompanied by lowered Ki67 expression and increased levels of Fas and LAG-3. The observed contrast may highlight differences between naïve and memory $T_{CD8}$ in terms of their quality and threshold of activation by bacterial SAgs and before they are exposed or re-exposed to cognate IAV Ags, respectively. Therefore, whether the immune system sees an entirely new IAV strain or IAVs harboring similar Ags to those previously seen may determine, at least partially, how $T_{CD8}$ will behave when they encounter a SAg. This has obvious implications for pandemic and seasonal flu with complications arising from sepsis, non-menstrual toxic shock, food poisoning and other SAg-mediated illnesses and conditions.

Huang *et al.* previously reported that exposure to SEB seven days after a primary infection with X31 does not change the frequency of TCR $Vβ8.3^+$ cells among $NP_{366-374}$-specific $T_{CD8}$ (an immunodominant population in C57BL/6 mice) upon a PR8 challenge [80]. In contrast with our recall system, Huang *et al.* examined $NP_{366-374}$-specific $T_{CD8}$ within the BAL of intranasally primed-boosted mice. MHC II molecules in the C57BL/6 strain have a notoriously low affinity for SEB, which may explain why a larger dose of SEB (100 μg) had to be administered. Therefore, even though $NP_{366-374}$-specific $T_{CD8}$ preferentially use Vβ8.3, they may not establish strong and/or long enough contacts with SEB and MHC $II^+$ APCs.

In our initial PR8 infection model (Fig 7), SEB was injected i.p. first to mimic systemic exposure to this SAg. SEB administered through this injection route can reportedly find its way into the lungs [68]. We found the absolute numbers of anti-IAV $T_{CD8}$ to remain unaltered in the lungs and within the BAL. In contrast, there was a significant rise in the splenic $NP_{147}$-specific response. According to a mathematical model, the spleen contributes the lion's share of the overall IAV-specific $T_{CD8}$ response in the lungs, especially at later time points post-IAV infection [81]. However, it was pertinent to find out whether intranasal co-administration of SEB and PR8 may affect the local PR8-specific response. Indeed, we observed an increased $NP_{147}$-specific response in the lungs of SEB-pretreated mice in this setting (Fig 8), a finding that could be recapitulated in a superinfection model (Fig 9).

The bacterial culprits of SAg-mediated illnesses typically express more than one toxin. How multiple SAgs encoded by a single bacterial species may cross-regulate T cell responses to each other is unknown. Furthermore, we previously demonstrated that SAg-provoked T cell activation is modulated by Toll-like receptor 2 (TLR2) ligands embedded within the cell wall peptidoglycan of *S. aureus* [39]. Therefore, the effects of SAgs on anti-IAV immunity may also vary

depending on the presence, absence or physical proximity of the very bacteria that secrete these toxins. In our superinfection model, infection with an SEB-deficient mutant of *S. aureus* COL resulted in a weaker $T_{CD8}$ response to $NP_{147}$ (Fig 9). Therefore, SEB production by COL enhanced the subsequent response to this IAV epitope, an effect that could have been potentially more pronounced in the absence of the inhibitory effects of TLR2 agonists. The implications of this finding may be clinically relevant as far as secondary viral infections in the aftermath of toxic shock and sepsis are concerned.

Another important question is whether exposure to bacterial SAgs similarly modulates cognate $T_{CD8}$ responses to other viruses, especially those used in vaccination modalities. We addressed this question in a VacV immunization model in a pilot study. We chose to work with VacV because: i) it is very different from IAV structurally and genetically and also in terms of tissue tropism; ii) it is used in prophylactic vaccination against smallpox; iii) poxviral vectors have been employed in vaccine formulations to protect against other infectious diseases, including influenza [82,83,84]; iv) several VacV-derived epitopes have been identified in BALB/c mice [85], which enables studies on $T_{CD8}$ immunodominance.

Intriguingly, SEB administration before immunization with VacV failed to increase $T_{CD8}$ responses to this virus (S4 Fig). In fact, both local (peritoneal) and systemic (splenic) responses against $F2L_{26}$, the immunodominant peptide epitope of VacV in BALB/c mice [85] were dramatically reduced. By comparison, $T_{CD8}$ responses to two subdominant determinants ($A52R_{75}$ and $E3L_{140}$), were preserved in SEB-pretreated animals (S4A–S4D Fig). Why the immunodominant $T_{CD8}$ response to VacV is depressed, rather than elevated, is unknown at this point and a subject of our ongoing investigation. Unlike IAV, VacV readily propagates in the peritoneal cavity to establish an active infection. However, the ability of a viral intruder to propagate locally should not account, at least largely, for the observed differences. Our finding that SEB could enhance the splenic $NP_{147}$-specific response following both i.p. and i.n. IAV inoculations (Fig 1 and Fig 7C) lends support to the above notion. This is because unlike the i.p. injection of IAV, its i.n. administration results in viral propagation and active infection. At this point, we favor the possibility that distinct requirements for $T_{CD4}$ help, APC types and/or $T_{CD8}$ priming pathways may determine the susceptibility of antiviral $T_{CD8}$ to SAgs. In our IAV infection experiments, SEB reduced virus-specific IgG2b titres in the serum (S5 Fig), which can be viewed as an indirect measure of $T_{CD4}$ help in Ab class-switching. However, despite the apparent SEB-induced anergy in the $T_{CD4}$ compartment, the splenic (systemic) $NP_{147}$-specific $T_{CD8}$ frequency was raised (Fig 7C). In contrast, primary anti-VacV $T_{CD8}$ responses are known to depend on $T_{CD4}$ help when the virions are injected via the i.p. route [86]. How various SAgs may impact APC functions, for instance after binding to MHC II, and potentially promote one priming pathway versus another in different organs remains an open question.

In summary, we have investigated how bacterial SAgs, including but not limited to SEB, alter both the strength and the breadth of $T_{CD8}$ responses to multiple IAV strains. In doing so, we employed several *in vivo* models of primary and recall anti-IAV immunization and active viral infection. We propose that the nature and the TCR makeup of T cell populations participating in IAV-specific responses, the phase at which a SAg (or multiple SAgs) is/are encountered, and the means and modes of IAV Ag processing and presentation are among the most important determinants of protection versus immunopathogenesis in the context of vaccination and superinfections with IAV and Gram-positive bacterial pathogens.

## Supporting information

**S1 Fig. SEB administration before PR8 immunization increases the clonal size of $HA_{518}$-specific $T_{CD8}$.** BALB/c mice were injected i.p. with PBS or with 50 µg of SEB three days before

they were immunized i.p. with the PR8 strain of IAV. Seven days later, $HA_{518}$-specfic $T_{CD8}$ were identified through surface staining with an anti-CD8α mAb and MHC I dextramers. Representative dot plots after live gating on $CD3^+$ events are demonstrated in panel A. In addition, the absolute numbers of splenic $HA_{518}$-specific cells were calculated (B). Data are shown as mean ± SEM for 4 mice per group. ** denotes a statistically significant difference with $p<0.01$ using an unpaired Student's *t*-test.
(TIF)

**S2 Fig. Pretreatment with SEB prior to intramuscular immunization with heat-inactivated PR8 invigorate $NP_{147}$-specific $T_{CD8}$.** (A) Mice were injected i.p. with PBS or with 50 μg SEB followed, three days later, by i.m. vaccination with heat-inactivated PR8. Seven days after vaccination, the frequency (B,C) and the absolute number (C) of $NP_{147}$-specific $T_{CD8}$ was determined in each spleen by ICS for IFN-γ. Representative dot plots (B) and summary data (C) are depicted. Error bars represent SEM, and ** and *** denote statistically significant differences with $p<0.01$ and $p<0.001$, respectively, using an unpaired Student's *t*-test.
(TIF)

**S3 Fig. Pretreatment with SEB selectively enhances primary $NP_{147}$- and $HA_{518}$-specific $T_{CD8}$ responses elicited by SEQ12 immunization.** (A) BALB/c mice (n = 4/group) were injected i.p. with PBS or with 50 μg SEB three days before they were immunized i.p. with SEQ12. At the peak of the primary response (*i.e.*, on day 7 post-immunization), $T_{CD8}$ recognizing the indicated epitopes were enumerated by MHC I tetramer staining (B) and by ICS for IFN-γ (C). Error bars represent SEM. * and ** denote statistically significant differences with $p<0.05$ and $p<0.01$, respectively, which were determined using an unpaired Student's *t*-test
(TIF)

**S4 Fig. SEB administration prior to VacV immunization attenuates local and systemic $T_{CD8}$ responses to the immunodominant VacV-derived epitope.** (A) Mice were injected i.p. with PBS or with 50 μg SEB three days before they received VacV i.p. On day 6 post-immunization, a time point at which VacV-specific $T_{CD8}$ responses reach their peak, peritoneal (B,D) and splenic (C,E) responses to indicated peptides were quantified by ICS for IFN-γ. Representative zebra plots in B and C illustrate the frequencies of peritoneal (local) and splenic (systemic) $T_{CD8}$ responses to $F2L_{26}$, respectively. Summary data from PBS- and SEB-treated mice (n = 5/group pooled from two independent experiments) also depict subdominant responses to $A52R_{75}$ and $E3L_{140}$ (D,E). Error bars represent SEM, and * and ** denote significant differences with $p<0.05$ and $p<0.01$, respectively, using an unpaired Student's *t*-test.
(TIF)

**S5 Fig. SEB administration prior to a sublethal PR8 infection results in reduced antiviral IgG2b levels.** BALB/c mice were infected i.n. with 0.3 $MLD_{50}$ of PR8 three days after they received an i.p. injection of PBS or SEB (50 μg). Three weeks after PR8 infection, mice were terminally bled, and the presence of PR8-specific IgG2b was evaluated in serially diluted serum samples as described in Materials and Methods. Error bars represent SEM. *, ** and *** denote statistically significant differences with $p<0.05$, $p<0.01$ and $p<0.001$, respectively.
(TIF)

## Acknowledgments

We thank Delfina Mazzuca for preparation of bacterial superantigens, Tunyalux Langsub for her assistance with figures, and other members of the Haeryfar Laboratory for helpful discussions. We are grateful to Dr. Jonathan Yewdell (National Institutes of Health) for critically

reviewing our manuscript and to Dr. Suzanne Epstein (Food and Drug Administration) for her expert advice on detection of IAV-specific antibodies.

## Author Contributions

**Conceptualization:** Tina S. Mele, Bhagirath Singh, Jimmy D. Dikeakos, Hong-Hua Mu, Jack R. Bennink, John K. McCormick, S. M. Mansour Haeryfar.

**Data curation:** Courtney E. Meilleur, Arash Memarnejadian, Adil N. Shivji, John K. McCormick, S. M. Mansour Haeryfar.

**Formal analysis:** Courtney E. Meilleur, Arash Memarnejadian, Adil N. Shivji, John K. McCormick, S. M. Mansour Haeryfar.

**Funding acquisition:** S. M. Mansour Haeryfar.

**Investigation:** Courtney E. Meilleur, Arash Memarnejadian, Adil N. Shivji, Jenna M. Benoit, Tina S. Mele, Bhagirath Singh, Jimmy D. Dikeakos, John K. McCormick, S. M. Mansour Haeryfar.

**Methodology:** Courtney E. Meilleur, Arash Memarnejadian, Stephen W. Tuffs, Jack R. Bennink, John K. McCormick, S. M. Mansour Haeryfar.

**Project administration:** S. M. Mansour Haeryfar.

**Resources:** David J. Topham, Hong-Hua Mu, Jack R. Bennink, John K. McCormick, S. M. Mansour Haeryfar.

**Supervision:** S. M. Mansour Haeryfar.

**Validation:** Courtney E. Meilleur, S. M. Mansour Haeryfar.

**Writing – original draft:** Courtney E. Meilleur, S. M. Mansour Haeryfar.

**Writing – review & editing:** S. M. Mansour Haeryfar.

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
