## [Decision Letter · Decision Letter 0]

28 Aug 2019

Dear Dr. Haeryfar,

Thank you very much for submitting your manuscript "Discordant rearrangement of primary and anamnestic CD8 T cell responses to influenza A viral epitopes through systemic exposure to bacterial superantigens: implications for superinfections, prophylactic vaccination and heterosubtypic cross-protection" (PPATHOGENS-D-19-01248) for review by PLOS Pathogens. Your manuscript was fully evaluated at the editorial level and by independent peer reviewers. The reviewers appreciated the attention to an important problem, but raised some substantial concerns about the manuscript as it currently stands. These issues must be addressed before we would be willing to consider a revised version of your study. We cannot, of course, promise publication at that time.

We therefore ask you to modify the manuscript according to the review recommendations before we can consider your manuscript for acceptance. Your revisions should address the specific points made by each reviewer.

(1) A letter containing a detailed list of your responses to the review comments and a description of the changes you have made in the manuscript. Please note while forming your response, if your article is accepted, you may have the opportunity to make the peer review history publicly available. The record will include editor decision letters (with reviews) and your responses to reviewer comments. If eligible, we will contact you to opt in or out.

(2) Two versions of the manuscript: one with either highlights or tracked changes denoting where the text has been changed; the other a clean version (uploaded as the manuscript file).

Additionally, to enhance the reproducibility of your results, PLOS recommends that you deposit your laboratory protocols in protocols.io, where a protocol can be assigned its own identifier (DOI) such that it can be cited independently in the future. For instructions see http://journals.plos.org/plospathogens/s/submission-guidelines#loc-materials-and-methods

We hope to receive your revised manuscript within 60 days. If you anticipate any delay in its return, we ask that you let us know the expected resubmission date by replying to this email. Revised manuscripts received beyond 60 days may require evaluation and peer review similar to that applied to newly submitted manuscripts.

[LINK]

Sincerely,

Matthew S Miller, PhD

Guest Editor

PLOS Pathogens

Adolfo García-Sastre

Section Editor

PLOS Pathogens

Kasturi Haldar

Editor-in-Chief

PLOS Pathogens

orcid.org/0000-0001-5065-158X

Grant McFadden

Editor-in-Chief

PLOS Pathogens

orcid.org/0000-0002-2556-3526

While the reviewers largely appreciated the robustness of the presented data, significant and unanimous concerns were raised about the "artificial" nature of the experimental systems used. While the data it interesting, the reviewers commented that in its current form, it is mainly "phenomenology" that has questionable relevance for actual influenza virus/bacterial co-infections (or vaccinations).

The authors should repeat their experiments in systems which more closely mimics actual influenza virus vaccination and co-infection conditions/exposure routes. Specifically, inactivated influenza should be administered i.m. to mimic inactivated influenza virus vaccination (and/or an attenuated influenza virus could be administered i.n. to mimic FluMist). Secondly, in the context of both infection and vaccination, SEB should be administrated into the lungs at moderate levels that would mimic those predicted to occur during natural co-infection, or better still, an SEB-expressing pathogen could used to confirm the reported phenotypes under more physiologically-relevant conditions.

Reviewer's Responses to Questions

**Part I - Summary**

Reviewer #1: Influenza infection in the respiratory tract is known to greatly increase susceptibility to bacterial infection. Thus research to study the interaction between influenza infection and exposure to bacterial superantigens may be of interest in understanding how to improve clinical outcomes. However this paper studies something completely different. Mice who are naïve to influenza, and therefore representing only very young children, are exposed to SEB by the i.p. route, and then influenza virus by the i.p. route. This does not model any situation that occurs in humans.

The paper states that 'IAVs constantly mutate to escape TCD8-mediated immunosurveillance' without citing a reference. Where is the evidence that this is true? Antigenic drift definitely occurs in the surface glycoproteins but internal antigens are highly conserved. There is very little evidence for 'immunodominance' of influenza virus epitopes in humans.

The researchers go on to assess SEB administration before immunization with 'rVVs'. Presumably recombinant Vaccinia viruses, which need a much better description in the methods section - what strain, and how are the minimal epitopes targeted to the ER (and also, why?) Why use epitopes only instead of whole antigens?. Again, humans will never be immunized with rVVs expressing influenza epitopes. Humans will never be immunized by rVVs expressing whole influenza antigens either, and especially not by the i.p. route.

My issue with this manuscript is my failure to understand why the research was conducted.

Reviewer #2: This is an interesting study that explores how superantigens, particularly SEB, influence CD8 T cell responses to influenza A virus (IAV) infection. In well designed and well executed experiments they show that SEB injected three days before A/PR8 IAV infection enhances the T cell response to peptide epitopes, where SEB-reactive TCRs are part of the response. They extend this to other superantigens and also use a mutated SEB to show the enhanced response is superantigen specific. These results were all found when the infection was intraperitoneal and when SEB was given ip and the IAV intranasally, there was no difference in the lung T cell response, though the splenic response was enhanced. Following ip SEB and IAV priming, secondary responses to a second (T cell ) cross reacting virus were enhanced. However, when SEB was given after the first priming, the secondary T cell response was decreased, implying differential effects of the SEB on naïve versus memory T cells. However early exposure to SEB after priming, enhances the T cell response.

Overall the issue is of some importance given the well-known but poorly understood interactions between IAV and bacterial infections in humans, that have also been explored to some extent in mouse models. However this model using purified super antigens and IP IAV infection is rather oversimplified. When they infect intranasally the findings become quite complex. Overall it iis not clear how relevant these findings will be to humans and that could be discussed and reviewed in more detail in the Discussion.

Reviewer #3: In this study by Meilleur et al, the authors investigate the impact of bacterial superantigen SEB on CD8 T cell responses. Intraperitoneal administration of SEB followed by ip administration of the influenza virus PR8 strain increased the numbers of CD8 T cells. This increase was specific to T cells with Vb3 or Vb8 TCR. A similar increase in CD8 T cells was observed with several influenza virus strains when SEB was administered prior to influenza virus injection. However, SEB administration had no effect on the numbers of CD8 T cells in the respiratory tract upon intranasal administration of influenza virus. There was some increase in NP147 epitope specific CD8 T cells in the spleen. Interestingly, SEB administration in influenza virus primed mice prior to boosting resulted in significantly decreased CD8 T cell numbers. Overall, the authors convincingly demonstrate that SEB administration prior to influenza virus administration increases the number of CD8 T cells with Vb3 and Vb8 TCR. A major weakness in the study is the relevance of the model system used here - ip injection of infectious influenza virus is very different from intramuscular injection of inactivated vaccine or natural infection of the respiratory tract. Thus, the significance of these results in the context of co-infection or the impact of SEB on influenza virus specific CD8 T cell responses is not thoroughly addressed.

Reviewer #4: The manuscript addresses an interesting and medically important subject. The approach used to examine the effect of superantigen exposure on the response to IAV and/or IAV associated peptides in robust and should help gain a better understanding of the complex role superantigens and/or superinfections play in the response to influenza and possibly other viruses. Further, the findings of the importance of Vβ chain expression on the response to SEB exposure is novel. However, there are a number of issues that weaken the strength and impact of this work

**Part II – Major Issues: Key Experiments Required for Acceptance**

Reviewer #1: (No Response)

Reviewer #2: Rather oddly in the Discussion they immediately discuss results not mentioned in the Results section but shown only in the supplementary figures where rather different results were found using recombinant vaccinia or minigene priming. Then the normally immunodominant response to NP147 epitope was almost completely suppressed. This raises issues about use of T cell receptors and antigen presentation that ought to be explored experimentally in the Results section. They could either complete these experiments and put in the main section, strengthening the paper, or cut this additional data set out of this version and present those results later in a second paper, in which case they should put a caveat summarising these findings in this Discussion. The latter would be easier but the former would improve the paper.

Reviewer #3: Major comments:

1. The authors should use a relevant model system. Time course experiments should be performed with SEB administration at various times (0, 3, 7, 10 days) post-infection (in) or post vaccination (im) to determine the effects of SEB on naïve and activated /memory CD8 T cells.

2. The majority of super infections or co-infections occur in the respiratory tract. It will be important to establish if administration of SEB at moderate levels in the respiratory tract affects IAV specific CD8 T cell responses.

3. Supplementary Fig 2 & 3 are not mentioned in the results section but only in the discussion section.

Reviewer #4: 1. The data, in its current form, is largely observational and does not provide solid evidence of a mechanism to support the observed effects. The authors provide reasonable explanations but no data to confirm. Additional work to substantiate their suggested mechanisms would significantly strengthen this work.

2. The work described in figure 4 has the potential, from a clinical standpoint, to be the most important observation. The effect of SEB exposure on the response in the lungs speaks to the clinical relevance of this work. In the current form, the authors suggests that the fact that they see a response in the spleen but not the lungs suggests that SEB exposure works locally and not at peripheral sights. While this is an important observation, no work was presented to describe whether local exposure in the lungs would result in alterations in the immune response to IAV. This is a critical experiment and would significantly increase the strength of this work.

**Part III – Minor Issues: Editorial and Data Presentation Modifications**

Reviewer #1: (No Response)

Reviewer #2: None

Reviewer #3: Minor comments

1. Does the amount of purified SEB administered in mice reflect the levels of SEB produced during bacterial infection?

2. Does SEB administration followed by SEQ12 result in increased TCD8 numbers?

3. P17 “N23 (Asn23) should also mediate the reactivity of SEB with mouse Vβ8.3” should read “not mediate reactivity”?

4. Fig 2C: As compared to SEB, mutant SEB N23A does not show significant decrease in NP147. The results section should be modified accordingly.

5. P16 “the above results demonstrate that systemic exposure to SEB expands,

rather than deletes…” As both the SEB and PR8 are injected ip, this is just local exposure and not systemic exposure.

Reviewer #4: Minor Points:

1. The data presented in figure 1G demonstrating an increased response to NP147 loaded target cells is not convincing without inclusion of the control data. The authors should consider including the histograms from the naïve (PBS and SEB) mice.

PLOS authors have the option to publish the peer review history of their article (what does this mean?). If published, this will include your full peer review and any attached files.

Reviewer #1: No

Reviewer #2: No

Reviewer #3: No

Reviewer #4: No

---

## [Decision Letter · Decision Letter 1]

10 Feb 2020

Dear Dr. Haeryfar,

We are pleased to inform you that your manuscript 'Discordant rearrangement of primary and anamnestic CD8+ T cell responses to influenza A viral epitopes upon exposure to bacterial superantigens: implications for prophylactic vaccination, heterosubtypic immunity and superinfections' has been provisionally accepted for publication in PLOS Pathogens.

Before your manuscript can be formally accepted you will need to complete some formatting changes, which you will receive in a follow up email. A member of our team will be in touch within two working days with a set of requests.

Best regards,

Matthew S Miller, PhD

Guest Editor

PLOS Pathogens

Adolfo García-Sastre

Section Editor

PLOS Pathogens

Kasturi Haldar

Editor-in-Chief

PLOS Pathogens

orcid.org/0000-0001-5065-158X

Michael Malim

Editor-in-Chief

PLOS Pathogens

orcid.org/0000-0002-7699-2064

The authours went to great lengths to carefully and comprehensively address the concerns raised during the previous round of reviews. The manuscript has been substantially improved - presenting a much clearer story with potentially important clinical implications.

Reviewer Comments (if any, and for reference):

Reviewer's Responses to Questions

**Part I - Summary**

Reviewer #2: The manuscript has been greatly improved and clarified by the revisions, in response to all the original reviews. In particular they have followed the suggestion of this reviewer in removing the Vaccinia-NP mini gene experiment, just discussing the response to vaccinia itself and making a caveat in the discussion.

Reviewer #3: The authors have reasonably addressed all the concerns/

**Part II – Major Issues: Key Experiments Required for Acceptance**

Reviewer #2: (No Response)

Reviewer #3: N/A

**Part III – Minor Issues: Editorial and Data Presentation Modifications**

Reviewer #2: No further modifications are now needed.

Reviewer #3: N/A

PLOS authors have the option to publish the peer review history of their article (what does this mean?). If published, this will include your full peer review and any attached files.

Reviewer #2: No

Reviewer #3: No

---

## [Editor Report · Acceptance letter]

15 Apr 2020

Dear Dr. Haeryfar,

We are delighted to inform you that your manuscript, "Discordant rearrangement of primary and anamnestic CD8+ T cell responses to influenza A viral epitopes upon exposure to bacterial superantigens: implications for prophylactic vaccination, heterosubtypic immunity and superinfections," has been formally accepted for publication in PLOS Pathogens.

Best regards,

Kasturi Haldar

Editor-in-Chief

PLOS Pathogens

orcid.org/0000-0001-5065-158X

Michael Malim

Editor-in-Chief

PLOS Pathogens

orcid.org/0000-0002-7699-2064